# CAN SPEECH LLMS THINK WHILE LISTENING?

**Yi-Jen Shih[1,2]\*, Desh Raj[2]\*, Chunyang Wu[2]\*, Wei Zhou[2]\*, SK Bong[2], Yashesh Gaur[2], Jay Mahadeokar[2], Ozlem Kalinli[2], Michael L. Seltzer[2]**

[1]The University of Texas at Austin
[2]Meta Superintelligence Labs
yjshih@utexas.edu, rdesh26@gmail.com, cwu564@gmail.com,
zhowei@google.com

## ABSTRACT

Recent advances in speech large language models (speech LLMs) have enabled seamless spoken interactions, but these systems still struggle with complex reasoning tasks. Previously, chain-of-thought (CoT) prompting or fine-tuning has been to shown to significantly improve the reasoning abilities of text-based LLMs. In this work, we investigate the effect of CoT fine-tuning for multi-stream speech LLMs, demonstrating that reasoning in text space **improves the accuracy of speech LLMs by 2.4x**, on average, over a suite of spoken reasoning tasks. Beyond accuracy, the latency of the spoken response is a crucial factor for interacting with voice-based agents. Inspired by the human behavior of "thinking while listening," we propose methods to reduce the additional latency from reasoning by allowing the model to start reasoning before the user query has ended. To achieve this, we introduce an entropy-based metric, "question completeness," which acts as an indicator to guide the model on the optimal time to start reasoning. This method provides greater **control** over the accuracy-latency trade-off compared with heuristic-based approaches and, under equivalent latency conditions, yields a 4% accuracy gain on ARC-Easy. Finally, we use Direct Preference Optimization (DPO) on preference data created using rejection sampling to push the accuracy-latency pareto frontier further, resulting in a **70% reduction in latency** without loss in accuracy.

## 1 INTRODUCTION

The traditional approach for building voice agents is to cascade several components: an automatic speech recognition (ASR) model, a text-based large language model (LLM), and a text-to-speech (TTS) model Huang et al. (2023); Lin et al. (2024); Likhomanenko et al. (2025); Chen et al. (2025). The recent emergence of speech large language models (Speech LLMs) (Cui et al., 2024) offers a promising alternative to this cascaded pipeline. These models are designed to directly process speech input or generate speech output, thereby eliminating the need for separate ASR or TTS modules. This integrated approach can seamlessly process both the semantic content and paralinguistic features of speech, and also reduces latency due to cascaded components. In addition to being used for specialized tasks such as speech understanding (Tang et al., 2024; Hu et al., 2024; Lu et al., 2024) and speech generation (Ye et al., 2025; Du et al., 2024), these models have shown promise in end-to-end spoken dialog Défossez et al. (2024); Huang et al. (2025). Nevertheless, while they work well for casual conversations, speech LLMs often fall behind their text-based counterparts on complex reasoning tasks (Peng et al., 2025; Wei et al., 2025; Lin et al., 2025). Consequently, the development of methods to enhance the reasoning capabilities of speech LLMs remains an underexplored and critical research problem to further their application as smart companions.

Several approaches have been explored to enhance reasoning capabilities in text-based LLMs, with the most popular being chain-of-thought (CoT) (Wei et al., 2022). In this method, the model generates a series of intermediate reasoning steps before producing its final answer. An LLM can be coerced to elicit reasoning traces either through zero-shot prompting (Kojima et al., 2022) or via supervised fine-tuning on a dataset of reasoning examples (Zelikman et al., 2022). Despite their strong

---

\*Work done at Meta

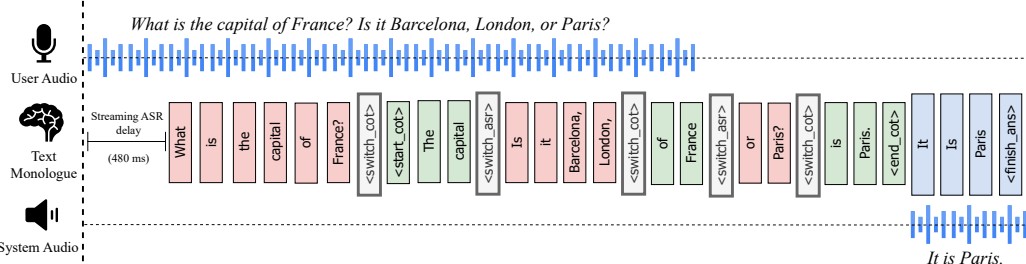

Figure 1: **Training token sequence arrangement**. We train the model to interleave reasoning tokens $\mathcal{R}^{\mathrm{T}}$ with streaming ASR tokens $\mathcal{Q}^{\mathrm{T}}$ on the text monologue channel, with special switch tokens for mode switching. After the CoT ends, the model generates text tokens which align with the spoken response $\mathcal{R}^{\mathrm{T}}$. For simplicity, [PAD] and [EPAD] tokens are not shown here.

performance, recent research highlights a critical trade-off with CoT: the length of the reasoning trace is a crucial factor in accuracy. While longer CoT sequences generally yield better performance, this comes at the cost of increased decoding latency and computational overhead (Jin et al., 2024). To tackle this problem, there are explorations about when and how long an LLM should reason (Sprague et al., 2025), resulting in a growing interest in "hybrid" reasoning models.

Although some recent work has adopted CoT in the speech domain, they focus primarily on applications such as speech translation Hu et al. (2025); Du et al. (2025); Gállego et al. (2025), dialogue Arora et al. (2025), or other detection tasks Mai et al. (2025); Park et al. (2025). The integration of CoT in speech LLMs requires answering two research questions: (i) should models reason using text or speech, and (ii) how do we maintain the responsiveness required for spoken interactions? To answer the first question, we investigate both alternatives, showing that text-based CoT is as performant as speech-based CoT for improving reasoning in speech LLMs, while being 2x more token-efficient.

The sequential process of listening, reasoning, and responding introduces considerable latency; consequently, previous research has proposed methods to overlap CoT tokens with speech to improve real-time conversational AI. Building upon the anthropomorphism of speech LLMs, concurrent works such as STITCH (Chiang et al., 2025) and Mini-Omni-Reasoner (Xie et al., 2025) have proposed "thinking while speaking," i.e., the model begins its spoken response while its reasoning is still ongoing. This is achieved by interleaving chunks of reasoning tokens with spoken response tokens, and subsequent CoT chunks are generated in the time it takes for the audio decoder to synthesize the preceding response. Despite showing reasonable improvements, this approach has notable limitations. For instance, the optimal chunk size for interleaving requires careful tuning and is dependent on hardware limitations. Moreover, despite a reduction in the time to first word, the model may inadvertently vocalize too much of its reasoning, leading to a longer overall response time to a final, conclusive answer. In this paper, we draw inspiration from neuroscience (Donhauser & Baillet, 2019) to propose a novel "thinking while listening" paradigm, by enabling concurrent processing of text-based CoT and user speech.

Current speech LLM architectures may be broadly categorized into two types: single-stream and multi-stream. Single-stream architectures merge user/system speech and text into a unified token sequence (Kim et al., 2024; Veluri et al., 2024), while multi-stream architectures simultaneously model distinct streams for each token sequence (Défossez et al., 2024). In this work, we build upon a multi-stream architecture due to its superior capacity for the concurrent processing of user audio and reasoning tokens. This design provides significant flexibility by allowing the system's text stream to be revised independently, a key advantage over single-stream models that lack this decoupling. Specifically, we fine-tune the publicly available Moshi model (Défossez et al., 2024) to generate CoT within its text monologue stream to improve its reasoning capabilities (Section 2). To enable the model to think while listening, we propose two methods: (i) a novel metric that estimates the completeness of the user's question at each timestep, and (ii) a preference tuning scheme to update the model's reasoning dynamically with new input (Section 3).

Since there are no existing standard reasoning evaluations for speech LLMs, we curated a suite of single-turn spoken reasoning tasks from well-known text-based reasoning benchmarks compris-

ing mathematical reasoning, social/physical interactions, and other general reasoning tasks (Section 4.2). Overall, our contributions are summarized below.

1. **Text-based CoT improves reasoning in speech LLMs.** To the best of our knowledge, we are the first to explore text-based CoT fine-tuning on multi-stream speech LLMs. Our method obtains 2.4x improvement in accuracy, on average, over the Moshi baseline across the SRQA tasks.

2. **Thinking while listening reduces reasoning latency.** We demonstrate that auto-regressive models that can generate tokens in sync with streaming user input can be taught to "think early" using entropy-based selection of trigger points. We achieve this using a novel Question Completeness metric that results in more controllable accuracy-latency trade-offs.

3. **Preference tuning enables adaptive reasoning for early-CoT models.** We use rejection sampling to curate correctness-based and length-based preference data and use them for DPO training, pushing the accuracy-latency pareto further and reducing $\sim$70% latency without loss in accuracy.

## 2 MULTI-STREAM SPEECH LLMS WITH CHAIN-OF-THOUGHT

### 2.1 BACKGROUND: MOSHI

Moshi (Défossez et al., 2024) is a full-duplex multi-stream model that simultaneously processes three distinct token streams at each timestep: user audio, system audio, and system text (referred to as the "text monologue"). For the audio streams, a separate codec model, Mimi, is used to encode audio waveforms into discrete tokens and back, operating at a frame rate of 12.5 Hz with 8 codebooks. Eventually, all three streams of inputs are represented as tokens:

$$\text{User Audio} : \mathbf{A}^{\text{U}} \in \{1, \dots, N_A\}^{L \times 8} \tag{1}$$

$$\text{System Audio} : \mathbf{A}^{\text{S}} \in \{1, \dots, N_A\}^{L \times 8} \tag{2}$$

$$\text{System Text} : \mathbf{T}^{\text{S}} \in \{1, \dots, N_T\}^{L}, \tag{3}$$

where $N_A = 2048$ is the size of each Mimi code-book and $N_T = 32000$ is the text vocabulary size. All streams have $L$ time-aligned tokens; text tokens are interleaved with padding tokens (`[PAD]` and `[EPAD]`[1]) to align with the corresponding audio tokens. Notably, since such aligned text token sequences are significantly shorter than the corresponding speech, the majority of text tokens in Moshi are simply padding tokens. The model architecture consists of a temporal transformer and a depth transformer, trained jointly using Negative Log Likelihood(NLL) loss. At each timestep $t$, the temporal transformer consumes $\mathbf{A}_t^{\text{U}}$ and $\mathbf{A}_t^{\text{S}}$, and predicts $\mathbf{T}_{t+1}^{\text{S}}$[2]. This token is fed into the depth transformer, which generates $\mathbf{A}_{t+1}^{\text{S}}$. The model is trained to estimate the following probability:

$$p\left(\mathbf{A}_{t+1}^{\text{S}}, \mathbf{T}_{t+1}^{\text{S}} | \mathbf{A}_{\leq t}^{\text{S}}, \mathbf{T}_{\leq t}^{\text{S}}, \mathbf{A}_{\leq t}^{\text{U}}\right). \tag{4}$$

Moshi was trained in multiple stages: (i) pre-training a text backbone LLM (Helium) using next token prediction, followed by (ii) post-training and fine-tuning with audio token sequences as well as multi-stream data, and (iii) fine-tuning with user-system dialogue data. Subsequent studies also showed the effectiveness of alignment training with direct preference optimization (DPO) to improve aspects such as factuality and safety (Wu et al., 2025). During inference, Moshi consumes user tokens and generates a system text and audio token per timestep. Due to the nature of the training sequences, we can force Moshi to generate a response by inserting an `[EPAD]` token on the text monologue stream. Conversely, inserting a `[PAD]` token forces it to remain silent.

### 2.2 FINETUNING WITH COT

Given a user's spoken question, $\mathcal{Q}^A$, our model predicts a spoken answer, $\mathcal{A}^A$, guided by a reasoning trace, $\mathcal{R}$. In the base Moshi model, all spoken responses $\mathcal{A}^A$ are preceded by aligned text $\mathcal{A}^T$ on the text monologue stream. To integrate CoT in this framework, we allow the model to additionally generate text-only reasoning tokens $\mathcal{R}^T$ without corresponding audio. Since both $\mathcal{R}^T$ and $\mathcal{A}^T$ are

---

[1] `[EPAD]` is used for indicating end of consecutive pad tokens.

[2] For simplicity, we neglect the delay pattern of first audio codebook and system text in our notation.

Table 1: Examples of questions where it is feasible to start reasoning early without impacting the correctness of the answer.

| # | Question | Reasoning | Answer |
|---|----------|-----------|--------|
| 1 | What is the capital of France ... *is it New York or Paris?* | The capital of France is Paris. | It's Paris. |
| 2 | If you flip a fair coin three times and get heads each time ... *what is the probability the fourth flip is heads?* | It's a fair coin, so probability of heads/tails is always 0.5. | The probability is 0.5. |

generated on the text monologue channel, we demarcate them using special `<start_cot>` and `<end_cot>` tokens, as shown in Fig. 1.

To help the model learn the relationship between the user's spoken question and the CoT, we also introduce a streaming ASR component into the text monologue, with the corresponding tokens denoted by $\mathcal{Q}^{\mathcal{T}}$ (red tokens in Fig. 1). Previously, Arora et al. (2025) and Yuen et al. (2024) have suggested using the user's audio transcript as an intermediate step in the CoT process for speech LLMs, but they focused on offline ASR. In contrast, our model naturally learns streaming ASR through word-aligned user transcripts right-shifted by $k$ tokens for look-ahead. Based on our preliminary experiments, we used $k = 6$ (equivalent to a 480 ms look-ahead), which was found to provide a good balance between latency and word error rate (WER). Finally, the text monologue contains the user transcript $\mathcal{Q}^T$, the reasoning $\mathcal{R}^T$, and the response text $\mathcal{A}^T$. To ensure all three streams—user audio ($\mathbf{A}^U$), system audio ($\mathbf{A}^S$), and text monologue ($\mathbf{A}^U$)—have the same length, we insert silent audio tokens as needed.

We fine-tune Moshi using the same NLL loss for next token prediction during the SFT stage, and use DPO loss for off-policy preference tuning. During inference, we apply force-decoding to our fine-tuned model and the baselines in two ways: (i) at the start of the user's question, we force-decode $k$ `[PAD]` tokens to accommodate the streaming ASR, and (ii) we force-decode the `<start_cot>` token after the user's question ends if the model has not generated it already.

## 3 THINKING WHILE LISTENING

Since our text-based CoT does not generate any aligned audio, naively inserting it between the query and the response can increase the perceived latency, thus impacting the naturalness of the human-system interaction. To alleviate this issue, our objective is to reduce this additional latency by mimicking the common cognitive trait wherein humans begin processing and reasoning before a question is fully articulated.

There are two scenarios where a model can begin reasoning early and yet provide the correct answer, as illustrated in Table 1. The first scenario includes questions which can be considered "complete" before reaching the end. In such cases, the model can start reasoning early and simply ignore the remaining question. In the second scenario, sufficient information may be available to start reasoning before the question ends, but the model still needs the remaining information to provide a correct response. We propose two different methods to enable early thinking. To endow the model with the ability for early reasoning, we created training examples by using our proposed Question Completeness metric. This metric is designed to identify the optimal time for the model to begin generating its CoT. Subsequently, we fine-tuned the model on this dataset to teach it to follow the distribution of these early-reasoning examples. Finally, we apply preference tuning to further enhance the performance of the model under early thinking scenario.

### 3.1 MEASURING THE QUESTION COMPLETENESS

Let us define the *inflection point* of a question as the timestep where sufficient information is available to begin reasoning. Our objective is to teach the model to identify such points in order to start its reasoning trace. A naive approach to identify the inflection point may be to shift the reasoning trace by a fixed number of frames or words, based on the heuristic that sufficient information is typically available a few words before the question concludes. However, this method is fundamentally limited by its lack of semantic awareness. For instance, in the query "What is the capital of

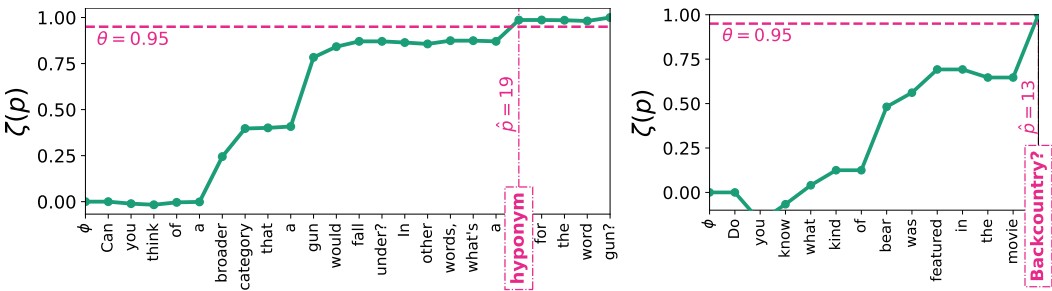

Figure 2: **Examples of the Question Completeness curve** $\zeta(p)$. In the first example, $\zeta$ reaches a high value at the end of the main question, at which point it is feasible to begin reasoning. In the second example, the word "Backcountry?" is critical to answer the question, and this is reflected in the corresponding $\zeta$ curve. More examples of the $\zeta$ curve are provided in Appendix A.1.

France?", a model cannot reasonably begin its reasoning process until the final word, "France," has been received. Consequently, it is necessary to develop a method that instructs the model to initiate reasoning at the *appropriate* moment, informed by the semantics of the question. We do this through a novel metric, which we call Question Completeness, denoted as $\zeta$.

Given a training sample that contains the question $\mathbf{Q}_{1:N}$, the reasoning $\mathbf{R}$, and the answer $\mathbf{A}$, where $N$ denotes the number of words in the question. Our goal is to find the index $p$ that splits $\mathbf{Q}$ into two halves: $\mathbf{Q}_{1:p}$ and $\mathbf{Q}_{p+1:N}$ such that

$$\Pr[\mathbf{R}, \mathbf{A}|\mathbf{Q}_{1:p}] \approx \Pr[\mathbf{R}, \mathbf{A}|\mathbf{Q}_{1:N}]. \tag{5}$$

Let $\mathbf{X}_p$ denote the joint probability of $\mathbf{R}$ and $\mathbf{A}$ given a partial question until the $p$-th word, i.e., $\mathbf{X}_p = \Pr[\mathbf{R}, \mathbf{A}|\mathbf{Q}_{0:p}]$. In practice, $\mathbf{X}_p$ can be estimated using an external language model. We define Question Completeness, $\zeta$, as:

$$\zeta(p) = 1 - \frac{D_{\mathrm{KL}}(\mathbf{X}_N||\mathbf{X}_p)}{D_{\mathrm{KL}}(\mathbf{X}_N||\mathbf{X}_0)}, \tag{6}$$

where $D_{\mathrm{KL}}$ denotes the Kullback-Leibler (KL) divergence. Here, $\mathbf{X}_N$ and $\mathbf{X}_0$ represent the extreme cases where the full question and no question are given, respectively. By definition, $\zeta(0) = 0$ and $\zeta(N) = 1$, so we can regard $\zeta$ as a semantic completeness progress bar[3]. Figure 2 shows illustrative examples of the $\zeta$ curve, indicating that $\zeta$ can be a good proxy for the progressive semantic completeness of a question.

The inflection point for a training sample can be approximated using $\zeta$ by estimating $\hat{p}$ s.t.

$$\hat{p} = \min\{p : \zeta(p) \geq \theta\}, \tag{7}$$

where $\theta$ is a hyperparameter. Further discussion and illustrative examples can be found in Appendix A.1.

## 3.2 SFT Tuning for Early Reasoning

To train the model to perform early reasoning, we construct the training token sequences by shifting the reasoning text tokens to the inflection point (as shown in Fig 1). Since the first `<switch_cot>` token corresponds to this inflection point, the model learns to identify the optimal timing by maximizing the probability of predicting this switch token. During inference, the inflection point is not provided. To initiate the reasoning, the model samples the first `<switch_cot>` token conditioned on all past generated tokens and the currently available partial user question. Consequently, the model must internally predict the inflection point by assessing the evolving semantic content of the partial user query. The entire training methodology is designed to increase the probability of sampling the the first `<switch_cot>` token at the moment the model determines the input is sufficiently complete.

---

[3]$\zeta$ is not guaranteed to be non-decreasing, and in practice, there are small local fluctuations in probability $\mathbf{X}_p$ due to incomplete syntax. Nonetheless, the general trend of $\zeta$ is still increasing from 0 to 1.

Figure 3: **The framework for curating preference data for DPO.** We generate outputs from the SFT model ($\pi_{\text{ref}}$) by force-decoding `<start_cot>` early (e.g., before "on which river" is spoken). The preferred response ($y_w$) is the one where the model is able to adaptively generate a correct and shorter reasoning trace.

### 3.3 PREFERENCE TUNING

While our question completeness metric allows for the creation of training samples that enable early reasoning, we observed that the model struggles to learn the distribution effectively via SFT and is often unable to update its CoT in response to new information in the user channel. Additionally, the CoT in our training data may be excessively long for simple questions, indicating a considerable opportunity to shorten the reasoning trace. To solve these issues, we created contrastive reasoning pairs, $\mathcal{D} = \left\{ \left( x^{(i)}, y_w^{(i)}, y_l^{(i)} \right) \right\}_{i=1}^{N}$, using rejection sampling and preference-tuned the SFT model using direct preference optimization (DPO) Rafailov et al. (2023). Fig. 3 illustrates our framework for preparing the preference dataset.

For a subset of prompts in the SFT training data, we generate $K$ responses using an SFT model (fine-tuned with early CoT) where we force-decoded `<start_cot>` at $\zeta(p) = \theta$ completeness. From these generations, we select a preferred output $y_w$ and a rejected output $y_l$. To improve adaptive reasoning, the preference is based on the correctness of the spoken response; for latency reduction, it is based on the both reasoning length and correctness. Kang et al. (2025) and Hao et al. (2024) have explored other techniques to reduce the the length of reasoning trace in CoT-based models.

Once we have the preference pairs, training is then performed using the DPO objective:

$$\mathcal{L}_{\text{DPO}} \left( \pi_{\Theta}; \pi_{\text{ref}} \right) = -\mathbb{E}_{(x, y_w, y_l) \sim \mathcal{D}} \left[ \log \sigma \left( \beta \log \frac{\pi_{\Theta} \left( y_w | x \right)}{\pi_{\text{ref}} \left( y_w | x \right)} - \beta \log \frac{\pi_{\Theta} \left( y_l | x \right)}{\pi_{\text{ref}} \left( y_l | x \right)} \right) \right], \quad (8)$$

where $\pi_{\Theta}$ and $\pi_{\text{ref}}$ are the policy and reference model respectively, $\sigma$ indicates sigmoid function and $\beta$ is a hyperparameter. Practically, we initialize the policy model $\pi_{\Theta}$ and the reference model $\pi_{\text{ref}}$ with the same weights from an SFT model $\pi$, but freeze the reference model during DPO training.

Following Wu et al. (2025), we calculate the token sequence probabilities exclusively using the text monologue stream $\mathbf{T}^{\text{S}}$ for a more stable training process, rather than using the full policy probability from eq. 4. Additionally, we exclude user streaming ASR tokens $\mathcal{Q}^T$ from this calculation to better differentiate between the probabilities of $\pi \left( y_w | x \right)$ and $\pi \left( y_l | x \right)$. We also adopt length-normalized DPO Meng et al. (2024) and add the NLL loss on $y_w$ to further stabilize the training Xu et al. (2024). The overall loss is given as:

$$\mathcal{L}_{\text{pref}} = \mathcal{L}_{\text{DPO}} - \lambda \mathbb{E}_{(x, y_w) \sim \mathcal{D}} [\log \pi_{\Theta} \left( y_w | x \right)], \quad (9)$$

where $\lambda$ is a hyperparameter that balances the two objectives.

### 3.4 INTERLEAVED REASONING WITH STREAMING ASR

In Section 2, we proposed that training the model to generate streaming user text tokens improves its textual reasoning capability. Predicting user text poses a challenge when left-shifting the reasoning trace to occur before the user's question is finished, since the CoT tokens may overlap with the existing streaming user ASR token sequence. To address this issue, we introduce two special switching tokens, `<switch_cot>` and `<switch_asr>`, which enable the model to alternate between the two generation modes on the text monologue stream.

To prepare the interleaving pattern for training, we first insert the user's streaming ASR tokens on the text channel as usual. Then, we identify available blank spaces (`[PAD]` and `[EPAD]` tokens) and insert the CoT tokens into these spaces. Whenever a mode switch occurs, we prepend the

corresponding switching token. This approach preserves the time alignment between the user's streaming ASR tokens and the audio input. A detailed illustration of the token arrangement can be found in Figure 1.

# 4 EXPERIMENTAL SETUP

## 4.1 TRAINING

Supervised fine-tuning for CoT based on the proposed modeling scheme requires training samples $(\mathcal{Q}, \mathcal{R}, \mathcal{A})$, where $\mathcal{Q}$ and $\mathcal{A}$ are in spoken formats, while $\mathcal{R}$ is in text format. Since there are no suitable large-scale public spoken reasoning datasets available, we used text-based reasoning datasets for training by converting them into spoken format. Specifically, we used the CoT-Collection Kim et al. (2023) dataset as it contains samples from diverse sources along with reasoning traces, amounting to a total of 1.8M examples. The reasoning traces in this dataset are augmented by OpenAI Codex followed by some filtering to ensure quality.

Since the CoT-Collection was created for text LLM training, it is not readily applicable to voice-based models. For instance, several samples are instances of summarization problems containing long-form text, which may not be applicable to natural conversations. We performed careful curation to obtain a spoken-friendly training dataset from this source:

1. Remove all samples where $\mathcal{Q}$ contains more than 60 words, resulting in ∼690K samples.
2. Use an LLM to perform spoken-friendly rewriting for all questions, reasoning, and answers. The system prompt used for this rewriting is provided in Appendix A.6.
3. Convert the rewritten questions and answers into audio waveforms using an internal TTS engine that generates 24Khz mono audios.

We have provided fine-tuning hyper-parameters and other details in Appendix A.2.

## 4.2 EVALUATION

**Spoken reasoning question-answering (SRQA) benchmark.** We prepared a suite of spoken reasoning tasks from multiple domains, derived from popular text benchmarks: (i) AI2 Reasoning Challenge (ARC) (Clark et al., 2018), (ii) Physical Interaction QA (PIQA) (Bisk et al., 2019), (iii) Social Interaction QA (SIQA) (Sap et al., 2019), and (iv) Grade School Math (GSM8K) (Cobbe et al., 2021). For ARC, we prepared easy (ARC-E) and challenging (ARC-C) subsets, similar to previous work. Since these evaluation tasks are derived from text sources, we used the same method of LLM-rewriting and TTS as used for the Spoken CoT-Collection, to convert them into spoken forms. We designed customized rewriting prompts for each eval set to ensure that the rewritten questions and answers are reasonable. Since several of the tasks contain multiple-choice questions, these were rewritten such that the choices are listed in the spoken question. Additionally, we also tracked the accuracy on LLaMA-Questions (Nachmani et al., 2024) to measure the model's performance for cases where reasoning may not be useful. The statistics and illustrative examples for all evaluation datasets can be found in Appendix A.3.

**Accuracy Calculation.** Throughout this work, we used LLaMA-3.1 405B Grattafiori et al. (2024) as a text-based judge to assess the correctness of the response. Since the model generates a spoken response $\mathcal{A}$, we used Pyannote VAD (Bredin & Laurent, 2021; Bredin et al., 2020) to first detect speech presence, followed by Whisper (Radford et al., 2023) to transcribe the response. The LLM-judge is provided the question and ground-truth answer, along with the model's transcribed response. It first determines whether the model provided an answer and then evaluates its correctness. The system prompt for the judge can be found in Appendix A.6.

**Latency Calculation.** In our experiments, latency is defined as the time interval from the end of the user's question to the initiation of the model's spoken response. To measure such latency, we first feed the evaluation user audio (question) into our model and save the entire response as an audio file. Then we apply Pyannote VAD (mentioned in the prior Accuracy Calculation paragraph) on the resulting audio file. This tool directly measures the end-to-end latency in seconds. However, instead of reporting in seconds directly, we chose to report latency in terms of number of tokens

Table 2: **Performance of text and speech LLMs on the SRQA benchmark.** All models are roughly 7B but vary in sizes of pretraining data. Our proposed method significantly enhanced the reasoning abilities of Moshi baseline and got competitive results against other speech LLMs pretrained with much more pretraining data.

| Model | # of pretraining text tokens | Reasoning | | | | | Factuality |
|---|---|---|---|---|---|---|---|
| | | ARC-E | ARC-C | SIQA | PIQA | GSM8K | LLaMA-QS |
| *Text LLMs* | | | | | | | |
| Helium[†] | 2.1T | 79.6 | 55.9 | 51.0 | 79.4 | – | – |
| LLaMA2-7b-Chat | 2T | 63.7 | 47.1 | 13.4 | 25.8 | 29.4 | 70.6 |
| Gemma-7B-Instruct | 6T | 82.5 | 66.2 | 18.3 | 45.0 | 43.1 | 69.7 |
| *Speech LLMs* | | | | | | | |
| Qwen2-Audio-7B-Instruct | 2.4T | 59.1 | 42.4 | 21.9 | 24.5 | **18.1** | **64.7** |
| Kimi-Audio-7B-Instruct | 18T | **83.0** | **71.5** | 32.9 | 34.4 | 15.7 | 61.7 |
| Moshi (baseline) | 2.1T | 30.2 | 21.5 | 22.8 | 23.8 | 8.7 | 42.8 |
| Moshi + **CoT (ours)**♣ | 2.1T | 77.7 | 59.8 | **56.1** | **56.9** | 16.1 | 57.8 |
|     w/o Streaming User ASR | 2.1T | 55.8 | 44.0 | 50.1 | 46.3 | 12.2 | 59.9 |

[†] We include Helium since it is the backbone for Moshi, but the results are not directly comparable. Since the model is not publicly accessible, we report the metrics from Défossez et al. (2024), which measured accuracy by evaluating the log-likelihood over the set of given options.
♣ Contains streaming user ASR with a delay of 6 tokens.

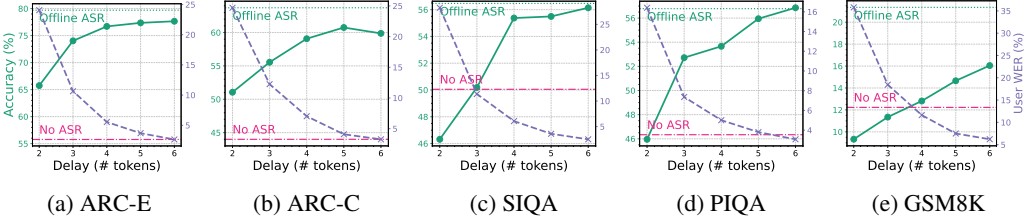

| (a) ARC-E | (b) ARC-C | (c) SIQA | (d) PIQA | (e) GSM8K |
|---|---|---|---|---|

Figure 4: **Effect of streaming user ASR on accuracy for SRQA tasks.** As we increase look-ahead, the accuracy improves and approaches the "offline ASR" topline.

since the 80ms[4] multiplier is configurable based on the speech tokenizer used, but the number of tokens actually represents how much the model's reasoning affects time to first packet.

Additionally, it is possible to further speed up the decoding of CoT text tokens depending on different implementations of model serving and hardware constraints. For example, we found that one forward pass for Moshi takes roughly 26ms on an A100 GPU. Hence, it is possible to decode 3 (80ms/26ms) text tokens along with 1 audio token to speed up the waiting time for CoT generation. Reporting latency in token length provides a hardware and implementation agnostic metric that ensures a fair and consistent comparison between our proposed model and the original Moshi baseline. We leave the implementation of this specific speedup strategy for future work.

## 5 RESULTS

### 5.1 CHAIN-OF-THOUGHT FINE-TUNING IMPROVES ACCURACY

**Comparison with baselines.** Table 2 shows the accuracy of our CoT fine-tuned model on the SRQA tasks, compared to the Moshi baseline. We also include several publicly available text and speech LLMs of similar size for reference (Touvron et al., 2023; Mesnard et al., 2024; Chu et al., 2024; Ding et al., 2025). On average, our proposed method provided an absolute accuracy improvement of 29.1%, with most eval tasks improving by 2-3x, showing the effectiveness of our approach. Among the speech LLMs, our fine-tuned model showed competitive performance across the board, placing top-2 on all reasoning tasks though the other speech LLMs are pretrained on much more pretraining data. The performance on LLaMA-QS also improved, but the gains were smaller.

**Effect of streaming user ASR.** In Section 2, we conjectured that training the model to transcribe the user's audio through aligned text tokens on the monologue stream would improve reasoning. To justify this choice, we designed an ablation experiment by removing these streaming user ASR tokens in training, but otherwise retaining the same training sequences. We also measured the topline accuracy using "offline" ASR, by training the model to transcribe the user audio after the end of

---

[4]For Mimi Codec, the duration of each audio token is 80ms.

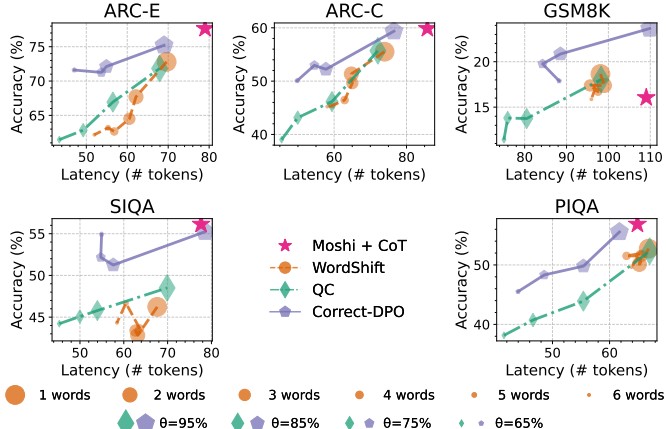

Table 4: **Effect of DPO training with length-based preferences** on accuracy (%) and latency (# tokens). The base SFT model is trained with $\theta = 0.75$. With DPO training, we further reduced latency by 70% without compromising the accuracy.

| Eval Set | Accuracy | | Latency | |
|---|---|---|---|---|
| | SFT | DPO | SFT | DPO |
| *LLaMA-QS* | 56.2 | 56.9 | 35.6 | 20.9 |
| ARC-E | 62.8 | 65.4 | 49.2 | 12.0 |
| ARC-C | 43.2 | 46.0 | 49.9 | 13.2 |
| SIQA | 45.1 | 45.3 | 50.0 | 12.9 |
| PIQA | 40.7 | 46.0 | 46.6 | 18.2 |
| GSM8K | 13.8 | 14.7 | 76.0 | 48.6 |

Figure 5: **Accuracy-latency curves for the proposed methods on SRQA reasoning tasks.** QC exhibits better controllability in trade-offs. DPO training with correctness-based preference further improves the accuracy of the QC models.

the question. Finally, we trained several models with streaming ASR where the user text tokens are delayed by different numbers of tokens (between 2 and 6). In Table 2, removing ASR significantly degrades accuracy on all reasoning tasks while remaining almost the same on factuality, corroborating that ASR helps reasoning. In Fig. 4, we observe that both user WER and accuracy consistently improved for SRQA tasks with an increasing number of delay tokens. The accuracy gain saturated after 4 delay tokens, though GSM8K showed continued improvement. At a delay of 6 tokens, the streaming ASR showed comparable accuracy to offline ASR for the majority of tasks.

**Reasoning in text v/s speech.** A design choice in our CoT fine-tuning is to perform reasoning in text, since text is much more information-dense than speech tokens. To investigate the effect of this decision, we conducted an ablation by performing CoT fine-tuning in speech. For this, we used our internal TTS engine to synthesize the CoT into speech and prefixed it to the spoken response.

For these ablation studies, we used the train/test subsets of GSM8K, and the results are shown in Table 3. As expected, the Moshi baseline performed poorly on GSM8K since it was trained primarily for casual dialog. CoT fine-tuning improved the overall accuracy from 8.7% to 17.5% and 17.2% for text-based and speech-based reasoning, respectively. While Speech CoT has zero latency but it comes with the cost of 3x token length for the entire response. Direct fine-tuning on question-answer pairs ("No CoT") degraded model performance, indicating that the improvement for CoT models cannot be attributed to our training data alone. Qualitative analysis revealed that in several cases where the Moshi baseline provided the correct answer, it actually performed some reasoning first. By directly fine-tuning it without CoT, we forced Moshi "not to think" and therefore reduced its accuracy. More results are shown in Appendix A.4.

Table 3: **Comparison of text-based and speech-based CoT** on GSM8K, in terms of accuracy (%).

| Model | Accuracy |
|---|---|
| LLaMA2-7b-Chat | 29.4 |
| Moshi (baseline) | 8.7 |
| Text CoT | 17.5 |
| Speech CoT | 17.2 |
| No CoT | 3.5 |

## 5.2 EFFECT OF EARLY REASONING

In Section 3, we proposed two methods to teach the model to think while listening: first, based on Question Completeness (QC), $\zeta$, and second, using DPO on reasoning traces generated with rejection sampling. For the QC method, we can control the onset of CoT during training based on $\theta$. As a simple baseline, we trained the model by left-shifting the CoT by a fixed number of words of the user question. We refer to this as WS-$N$, to denote shift by $N$ words. Fig. 5 shows the accuracy v/s latency curves for our proposed methods as well as the baseline. The latency metric is reported in terms of the number of tokens between the end of the user question and start of system response.

**QC-based shifting outperforms word-count heuristic.** First, it is evident that all latency improvements resulted in accuracy degradation, and different methods can only be compared based on their pareto-frontiers on the accuracy-latency curve. The results for the WS baselines were mixed: while

they showed gradual latency reduction on ARC, the performance on other tasks was haphazard. On PIQA and GSM8K, for instance, increasing $N$ in training did not result in expected reduction in latency, indicating that the model was unable to learn any patterns for early reasoning. The proposed QC method, on the other hand, provided better control over the trade-off. On all eval sets, latency improved as we reduced the $\theta$ for selecting inflection point (see equation 7) from 0.95 to 0.65.

**Correctness-based preference improves accuracy.** On further DPO training with correctness-based preference data, we achieved consistent improvements on all evaluation sets, as shown by the purple curve in Fig. 5. The marginal increase in latency stems from the better alignment between our model's behavior and the ground truth. We report the average gap between start CoT position of prediction and ground truth in Table 5. Negative sign indicates that the model generated CoT starts earlier than ground truth. Before applying Correct-DPO, SFT models tends to start CoT earlier than the ground truth.

Table 5: **Start CoT Gap** (# tokens) on the validation set. The gap is calculated by subtracting the position of <start_cot> between model generation and ground truth.

| $\theta$ | **Gap** (pred - gt) | |
|---|---|---|
| | SFT | Correct-DPO |
| 0.95 | -1.62 | -0.60 |
| 0.85 | -3.68 | -0.76 |
| 0.75 | -5.77 | -1.56 |
| 0.65 | -5.17 | -0.32 |

**Length-based preference improves latency**. Next, we trained the model using the length-based preference data to further shorten the CoT length. For this experiment, we chose the SFT model trained with $\theta = 0.75$ as our base model. From Table 4, we found that our method successfully reduced latency across all tasks by 30 tokens on average, while maintaining or improving accuracy.

Illustrative examples showing the improvements of our methods are shown in Appendix A.5.

## 6 CONCLUSION

In this work, we integrate CoT into speech LLMs and proposed a novel "thinking while listening" paradigm. We showed that text-based CoT can dramatically improve the model's performance, leading to 2-3x accuracy improvements over a suite of spoken reasoning tasks. A significant fraction of this improvement came from explicit user understanding by teaching the model to transcribe the user audio in a streaming manner. To avoid the latency cost of reasoning, we trained the multi-stream model to reason concurrently with incoming audio guided by our proposed "Question Completeness" metric. This metric offers a superior controllability over accuracy latency trade-off. We then performed preference tuning of this model with datasets created using rejection sampling. By curating correctness-based and length-based preference data, we were able to (i) improve the accuracy of the early-CoT model, and (ii) shorten the length of the reasoning trace, respectively. The resulting model demonstrated 70% lower latency without a significant loss in accuracy. Our work opens up a new direction for Speech LLMs: the ability to think while listening, a capability that brings them closer to the natural, responsive flow of human conversation.

## 7 ETHICS STATEMENT

We acknowledge that we have read and adhered to the ICLR Code of Ethics. Our research on Speech LLM reasoning is intended to advance the communication between human and machine. The data source used is publicly available and has no personally identifiable information.

## 8 REPRODUCIBILITY STATEMENT

To ensure the reproducibility of our experiments, we provide comprehensive implementation details in out paper. The overall pipeline for dataset preparation and evaluation is described in Section 4.1. Additionally, all LLM prompt templates and evaluation benchmark statistics are included in Appendices A.6 and A.3, respectively. The detail of our model training parameters is provided in Appendix A.2.

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

# A  APPENDIX

## A.1  THE "QUESTION COMPLETENESS" METRIC

In our preliminary experiments, we considered using entropy or log-probability as metrics for measuring completeness. However, both were found to be less robust, as they were more susceptible to noise from the incomplete syntax of partial questions. Consequently, we adopted Kullback–Leibler (KL) divergence for this purpose.

For determining the inflection point, we set a specific percentage-based threshold rather than capturing the largest jump in the curve, as suggested in prior work Labiausse et al. (2025). A large jump may occur early in a user's question when a key term is mentioned, but it does not mean that this partial information is sufficient to answer the question correctly. We conjecture that a metric based on a completeness percentage is more semantically reasonable. Furthermore, through manual examination of the QC curves on our training data, a 95% threshold was empirically found to align well with human perception of question completeness, serving as a conservative and effective criterion. We put more QC curves in the Appendix. Table 6 shows more examples of QC curves.

Table 6: Examples of the Question Completeness curve $\zeta(p)$. The word at inflection point $\hat{p}$ is shown as red and underlined. Each point on the horizontal axis corresponds to the cumulative sequence of words in the partial question up to and including the current word.

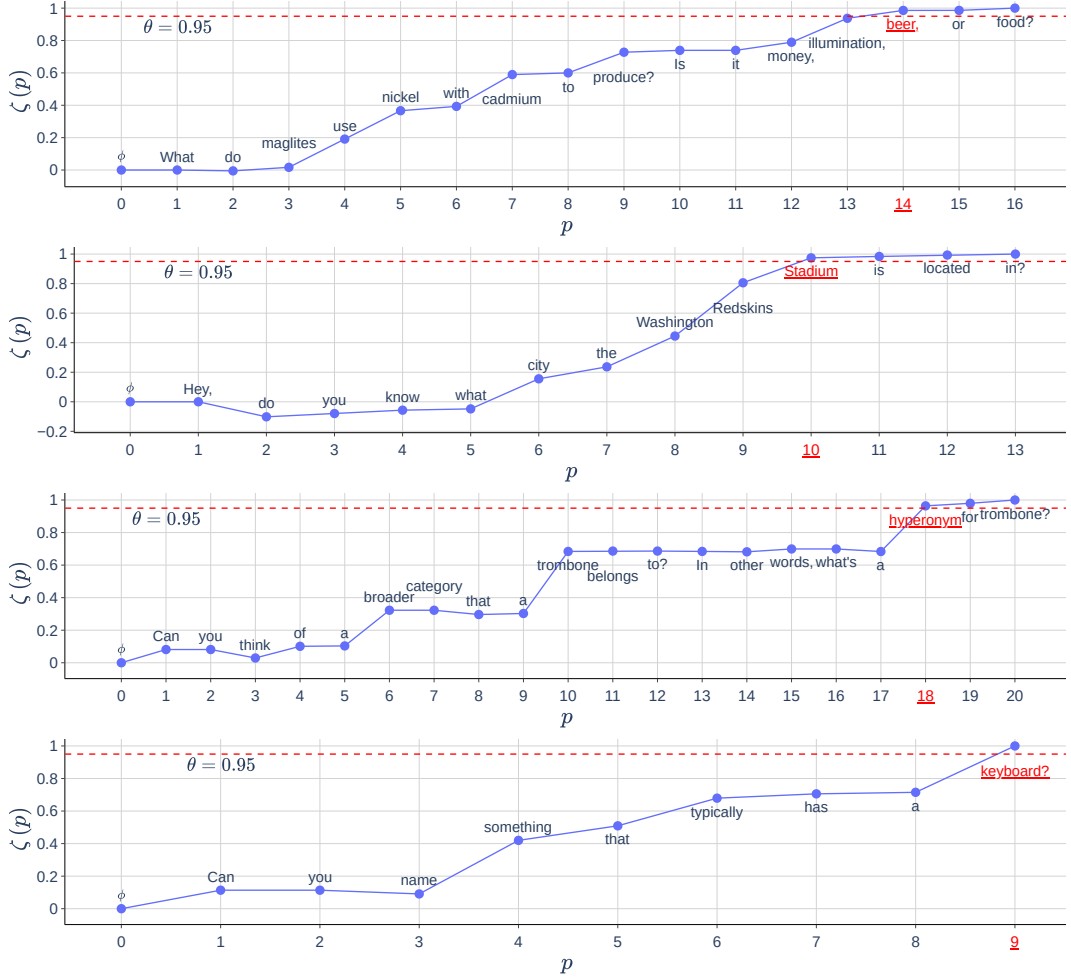

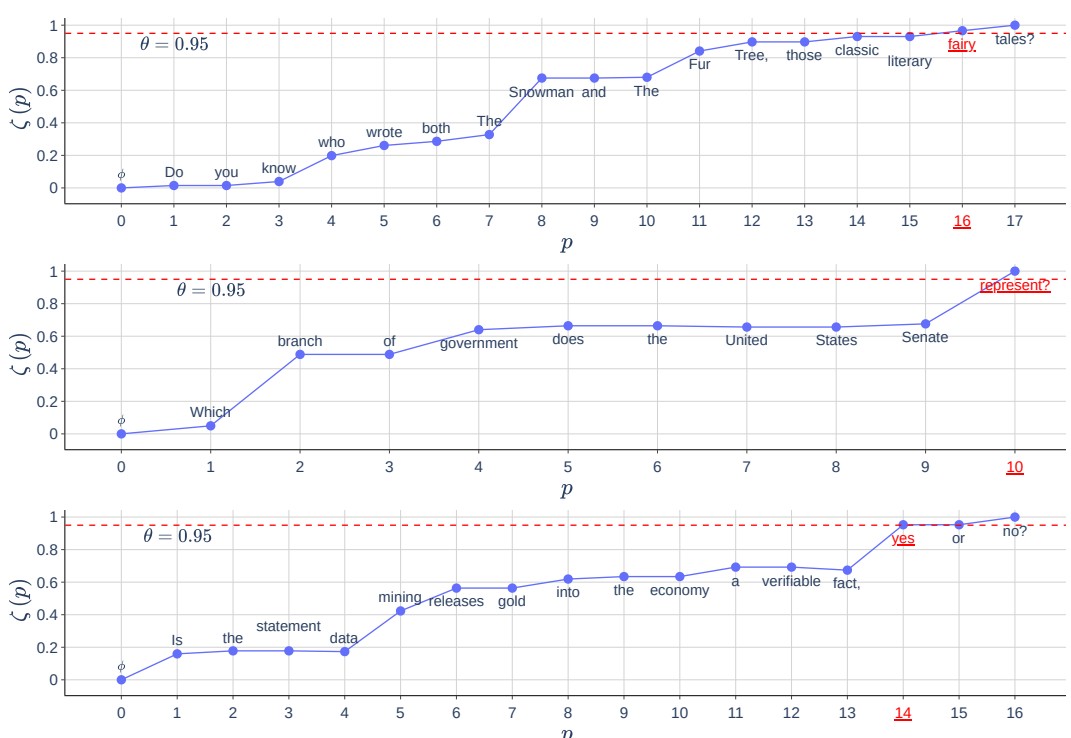

## A.2 TRAINING AND FINE-TUNING DETAILS

We fine-tuned the entire model with a learning rate (LR) of $4e-6$ and batch size 128 using fully-sharded data-parallel (FSDP) on 8 A100 GPUs. All models were trained for 8K steps with a warmup of 400 steps followed by LR annealing. We used Llama3-8B-Chat (Grattafiori et al., 2024) to estimate $\mathbf{X}_p$ which is required for estimating the inflection point $\hat{p}$ (§ 3.1). For preference tuning experiments, we selected models that are fine-tuned with different $\theta$ as the base models. We set learning rate to $5e-7$, $\beta = 0.1$, $\lambda = 0.1$, and trained with batch size 16 for 1200 steps. Final checkpoint was selected based on saturation of reward accuracy. To get a better monitor our model training, we curated a the validation set with a more strict filtering process. Specifically, we only keep examples with question length less than 80 words and the question shouldn't include keywords such as "paragraph", "article", ... etc. and no special character allowed. The rest of dataset preparation procedure is as same as the training set.

## A.3 SPOKEN REASONING BENCHMARK

Table 7 shows illustrative examples for each of the tasks in our SRQA benchmark. Since the source prompts for ARC-E, ARC-C, PIQA, and SIQA are choice-based tasks, LLM rewriting includes the vocalized options with the questions to make them suitable for spoken tasks.

Table 7: Statistics and illustrative examples for each task in the Spoken Reasoning Question Answering (SRQA) benchmark

| Task | Multiple choice | Size | $\mathcal{Q}$ statistics | | Example |
| | | | Dur. (s) | #words | |
|---|---|---|---|---|---|
| ARC-E | ✓ | 2376 | $14.5_{\pm 5.6}$ | $40.5_{\pm 17.5}$ | Q: Plants use sunlight to make something, but what is it? Is it soil, minerals, food, or water? *A: Food* |
| ARC-C | ✓ | 1172 | $16.9_{\pm 6.4}$ | $48.6_{\pm 19.6}$ | Q: What is the mass of a carbon atom that has 6 protons, 7 neutrons, and 6 electrons? Is it 6, 7, 13, or 19? *A: 13* |
| PIQA | ✓ | 1838 | $12.4_{\pm 4.7}$ | $43.9_{\pm 15.3}$ | Q: I want to install some cabinet pulls and I'm considering two options: either gluing some old jewelry under the cabinet knob or gluing it on top of the cabinet knob. Which do you think would be the better idea? *A: I think gluing the old jewelry on top of the cabinet knob would be the way to go.* |
| SIQA | ✓ | 1954 | $15.1_{\pm 3.7}$ | $49.8_{\pm 12.3}$ | Q: Hey, I was just watching this game and Ash had a pretty rough moment. He tried to redeem himself after missing an open shot. How do you think he's feeling right now? Would he be feeling disappointed, frugal, or maybe trying hard to shake it off? *A: I think he'd be feeling disappointed.* |
| GSM8K | ✗ | 1319 | $15.3_{\pm 5.7}$ | $46.3_{\pm 18.1}$ | Q: A candle melts by 2 centimeters every hour that it burns. How many centimeters shorter will a candle be after burning from 1:00 PM to 5:00 PM? *A: 8* |
| LLaMA-QS | ✗ | 300 | $3.0_{\pm 0.8}$ | $8.5_{\pm 2.1}$ | Q: Which river is the longest in South America? *A: Amazon* |

## A.4 COMPARISON OF TEXT-BASED AND SPEECH-BASED COT

In Table 3, we showed a comparison between text v/s speech modes of CoT, when trained and tested on GSM8K. While the accuracies were similar, reasoning in text was much more token-efficient than speech, with the token sequence lengths being $119.0 \pm 54.4$ and $368.8 \pm 190.3$, respectively. Speech-based CoT may still be useful as it introduces no additional latency compared to the baseline Moshi model, while still yielding substantial reasoning gains. This characteristic is similar to recent work such as STITCH (Chiang et al., 2025) that does "thinking while speaking". Still, text-based CoT results in shorter latency to the model's final answer since model generates the reasoning trace, without having to vocalize it.

Figure 6 shows the variation in accuracy for the different methods by the number of ground-truth reasoning steps required for the solution. For all models, accuracy degraded with an increase in # of steps, which indicates more complex questions.

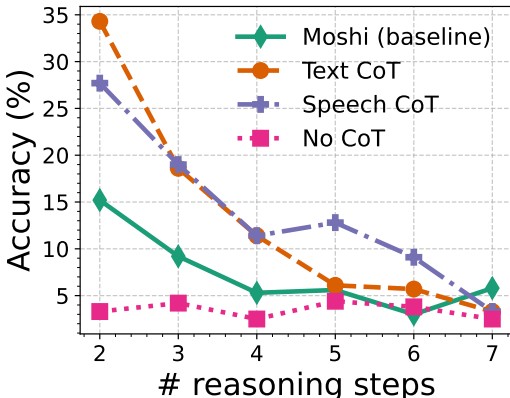

Figure 6: Accuracy variation by number of reasoning steps for text-based and speech-based CoT, measured on GSM8K.

## A.5 QUALITATIVE EXAMPLES

Table 8: Examples generated by models. We show some examples of the model output. Words in blue shaded are the spoken from the model. Words in red shaded are the user audio ASR. Words in green shaded are the CoT generated by the system.

---

**Example 1:**

| | |
|---|---|
| User Question: | Steve put together a puzzle that took 10 hours of hard work to complete. Anna put together the same puzzle in 2 hours more than half Steve's time. How long did it take Anna to finish the difficult puzzle? |
| Ground Truth | 7 |
| Moshi(Baseline): | Based on the information from the previous paragraph, we can infer that Anna took around six hours to complete the puzzle. |
| Moshi+CoT: | Steve put together a puzzle that took 1 0 hours of hard work to complete. Anna put together the same puzzle in two hours more than half Steve's time. How long did it take Anna to finish the difficult puzzle? If Steve's time to complete the puzzle is 1 0 hours, then half of this time is 1 0 / 2 = 5 hours. 2 hours more than half of Steve ' s time is 5 + 2 = 7 hours. Anna finished the puzzle in 7 hours. Anna took seven hours to finish the puzzle. |
| QC($\theta$=0.95) | Steve put together a puzzle that took 1 0 hours of hard work to complete. Anna put together the same puzzle in 2 hours more than half Steve's time. First find half of How long did it take Anna to finish the difficult the puzzle? time it took Steve to put together the puzzle : 1 0 hours / 2 = 5 hours. Then add 2 hours to find Anna's time : 5 hours + 2 hours = 7 hours. It took Anna seven hours to finish the puzzle. |

**Example 2:**

| | |
|---|---|
| User Question: | What would be best seen through a telescope? Would it be a moon, a cell, a light ray, or a molecule? |
| Ground Truth: | a moon |
| QC($\theta$=0.75)
+CorrectDPO | What would be best seen through a telescope? Would it be a moon, The telescope a cell, is a device that uses a light ray, lenses or a molecule? or mirror s to mag n ify images of distant objects. The most common use for telescope s is in astronomy, where they are used to observe celestial bodies such as planets and stars. A telescope can be used to see the moon because it's an object located in space. However, a cell cannot be seen through a telescope because cells are too small ( and therefore not visible ) without some sort of magnification device like a microscope. Light rays also do not need to be viewed through telescope s ; instead they are just part of the process by which we see things with our eyes. Mol ec ules don' t exist outside of chemical reactions so seeing them would require more than just looking at them through a lens. Therefore, " a moon " is the correct answer here you' d be able to see the moon through a telescope. |
| QC($\theta$=0.75)
+LengthDPO | What would be best seen through a telescope? Would it be a moon, The correct answer a cell, is A, because you can a light see ray, or the a molecule? moon through a telescope. You' d be able to see a moon through a telescope. |

**Example 3:**

| | |
|---|---|
| User Question: | Hey, I've got a friend named Jesse who just finished college after taking a few years to get his degree. Now that he's finally done, what do you think he'll want to do next? Will he want to enroll in college again, get a job next, or pay for college? |
| Ground Truth: | I think he'll want to get a job next. |
| QC($\theta$ = 0.75): | Hey, I' got a friend named Jesse who just finished college after taking a few years to get his degree. Now that he's finally done, what do you think he' ll want to do next? Here's the rationale Will he want to enroll in : " college again, river s flow trough get a job valley next, s." or pay for college? Actually, that's an unexpected answer, although it seems unrelated. However, according to the answer, rivers flow through valley s. |
| QC($\theta$=0.75)
+CorrectDPO | Hey, I' got a friend named Jessie who just finished college after taking a few years to get his degree. Now that he's finally done, what do you think he'll want to do next? Here's the rationale Will he want to enroll : in college again, " jj has finally get a job next, finished college, what or pay for college? will he want to do next? - he will probably want to get a job " I think he' ll probably want to get a job next. |

---

We present several qualitative examples generated by our models in Table 8. In Example 1, after fine-tuning with CoT, our model correctly answers the question, whereas the Moshi baseline fails. With our proposed QC-based early thinking, the model begins generating its CoT trace immediately after all information are provided. Therefore it reduces the latency.

In Example 2, we show an example requires minimal reasoning but the model generate a long CoT, which increase the latency a lot. By applying our Length-DPO fine-tuning, we were able to significantly reduce the CoT length while still maintaining the correct answer

Example 3 illustrates a limitation of the QC-based early thinking. If the model initiates reasoning too early—in this case, before the answer "get a job" is spoken—it is prone to generating an incorrect reasoning trace and, consequently, an incorrect final answer. With Correct-DPO tuning, the model overcome this failure. Even when the CoT trace starts at the same early point, the model correctly considers subsequent incoming information from the user question, leading to a correct answer.

A.6  SYSTEM PROMPTS

## System prompt for LLM-judge scoring

```
You are provided with a question, a ground truth answer and a model response. Your task is to determine
      whether the model response is correct.
Only determine the correctness of the response with the information provided.
Don't judge the non-factual components in the response, such as opinions, greetings, beliefs, subjective
      statements, follow-up questions.

Now Given

question: [{question}]

ground_truth_answer: [{gt_answer}]

model_response: [{model_output}]

Output should be a JSON-formatted string with dictionary containing keys (model_final_answer, judge_result).
      Do not include any other text.
For 'model_final_answer', please extract the final answer from the model_response.
If the model_response doesn't output a final answer, output '<no_final_answer>'.
If the model_response reaches a final conclusion, output the final answer (do not output any special
      characters).

For 'judge_result', please output one of the following three options:
1. output '<no_final_answer>' if the model_response doesn't conclude a final answer.
2. output '<correct>' if the model_final_answer is equivalent to the ground_truth_answer.
3. output '<incorrect>' if the model_final_answer is not equivalent to the ground_truth_answer.
Only output one of the above three options for 'judge_result'.
Plese judge it based on the only the given ground_truth_answer, the question and model_final_answer.

Example:
[Response]
{{"model_final_answer": "20", "judge_result": "<correct>"}}

[Response]
```

## System prompt for LLM rewrite on CoT-Collection

```
You are a helpful conversational assistant. Your task is to convert written question and answer pairs into a
      natural, spoken conversation. Do not throw away information required for answering the question. The
      question itself should be self-contained for people to answer it.
You are given a question an answer and a rationale. Please convert them into natural spoken conversation.
If it is a multiple choice question, please mention the choices in the converted spoken question.
If the given question is too long, please summarize it and include the information required for answering.
If the given quesiton refers to an article, passage, paragraph, please include the essential information in
      the converted question.
The converted_question and converted_answer should be in spoken format. The converted_rationale should be in
      written format (as concise as possible).
Do not use any special characters in the converted_question and converted_answer.
Make sure the converted_rationale is coherent with the converted_question and converted_answer.
The output shoud be in JSON format as the following.
{"converted_question": "...", "converted_answer": "...", "converted_rationale": "..."}

Example:

Question: [What was the reaction when the children were given ice cream?
Choose the most suitable option to answer the above question.
Options:

A. awesome

B. enjoyed

C. play chess

D. skip

E. sadness]]

Answer: [B]

Rationale: [The children were given ice cream, they enjoyed. So the answer is B]

Output:
{"converted_question": "What was the reaction when the children were given ice cream? Did they feel awesome,
      enjoyed, play chess, skip or sadness? Which one is more suitable", "converted_answer": ".They are most
      likely enjoyed.", "converted_rationale": "The children were given ice cream, they enjoyed. So the answer
      is enjoyed."}

Now given

Question: [{question}]

Answer: [{answer}]

Rationale: [{reasoning}]

Output:
```

## System prompt for LLM rewrite on SRQA benchmark

### ARC-E

```
You are a helpful assistant. Your task is to convert written question into a natural, spoken conversation. Do
    not throw away information required for answering the question. The question itself should be self-
    contained for people to answer it.
You are given a question, several options. Please convert them into natural spoken conversation. Make sure to
    mention the options in the converted spoken question.
The output should be in JSON format as the following.
{"converted_question": "..."}

Example1:

Question: [An astronomer observes that a planet rotates faster after a meteorite impact. Which is the most
    likely effect of this increase in rotation?]

Options: [1. Planetary density will decrease.
2. Planetary years will become longer.
3. Planetary days will become shorter.
4. Planetary gravity will become stronger.]

Output:
{"converted_question": "An astronomer observes that a planet rotates faster after a meteorite impact. Which is
     the most likely effect of this increase in rotation? Will planetary density decrease or planetary years
     become longer or planetary days become shorter or planetary gravity become stronger?"}

Now given

Question: [{question}]

Options: [{options_str}]

Output:
```

### ARC-C

```
You are a helpful assistant. Your task is to convert written question into a natural, spoken conversation. Do
    not throw away information required for answering the question. The question itself should be self-
    contained for people to answer it.
You are given a question, several options. Please convert them into natural spoken conversation. Make sure to
    mention the options in the converted spoken question.
The output should be in JSON format as the following.
{"converted_question": "..."}

Example1:

Question: [An astronomer observes that a planet rotates faster after a meteorite impact. Which is the most
    likely effect of this increase in rotation?]

Options: [1. Planetary density will decrease.
2. Planetary years will become longer.
3. Planetary days will become shorter.
4. Planetary gravity will become stronger.]

Output:
{"converted_question": "An astronomer observes that a planet rotates faster after a meteorite impact. Which is
     the most likely effect of this increase in rotation? Will planetary density decrease or planetary years
     become longer or planetary days become shorter or planetary gravity become stronger?"}

Now given

Question: [{question}]

Options: [{options_str}]

Output:
```

### PIQA

```
You are a helpful assistant. Your task is to convert written goal and solution into a natural, spoken
    conversation. Do not throw away information required for answering the question. The question itself
    should be self-contained for people to answer it.
You are given a goal, 2 solution and an answer. Please convert them into natural spoken conversation. Make
    sure to mention the options in the converted spoken goal.
The output should be in JSON format as the following.
{"converted_goal": "...", "converted_answer": "..."}

Now given

Goal: [{goal}]

Solution 1: [{option1}]
Solution 2: [{option2}]

Answer: [{answer}]

Output:
```

### SIQA

You are a helpful assistant. Your task is to convert written question into a natural, spoken conversation. Do
    not throw away information required for answering the question. The question itself should be self-
    contained for people to answer it.
You are given a question, 3 options and an answer. Please convert them into natural spoken conversation. Make
    sure to mention the options in the converted spoken goal.
The output should be in JSON format as the following.
{"converted_goal": "...", "converted_answer": "..."}

Now given

Goal: [{context} {question}]

Option 1: [{option1}]
Option 2: [{option2}]
Option 3: [{option3}]

Answer: [{answer}]

Output:

