# OpenReview forum: "Can Speech LLMs Think while Listening?"
_ICLR.cc/2026/Conference — ICLR 2026 Poster_

### Official Review · Reviewer_d8Do · 2025-10-24

**Soundness:** 2
**Presentation:** 2
**Contribution:** 4
**Rating:** 4
**Confidence:** 3

**Summary:**

This paper presents a method to interleave chain of thought (or any reasoning approaches) into speech language models. The naive way is to just perform CoT in text, before projecting it back to speech. This is a naive baseline that works, but increases latency.

The paper proposes a method to efficiently reason while the model is still listening to audio tokens. The method first uses something called a question-completeness metric that indicates when there is sufficient information in the question stream for the model to begin answering/reasoning. This metric is calculated using a model's output logits. The authors claim this allows us to generate a training dataset that we can perform SFT on the model.

Then, the authors introduced a preference optimization to further improves the model training.

**Strengths:**

I think the paper's overall idea is very relevant and practical for speech-based language models. In its introduction, it gives a clear outline about what the approach is about. I quickly could understand it is trying to interleave reasoning early on while the language model is still listening to the input streams.

The question-completeness metric also makes perfect sense, allowing us to identify when the LLM has enough information to start thinking about the answer (instead of listening till the end). Hence, I mark the paper's contribution as "excellent"

However, the paper leaves out the details on how exactly we can use their findings to fine-tune the LLM (e.g., how to use the offline training data created to actually perform SFT, and how that actually helps the LLM to determine the inflection point during runtime). I have some unanswered questions regarding this part, which is arguable the most important part of the paper.

**Weaknesses:**

While the paper does well in explaining the big idea, it has a significant weakness in explaining the details of its approach: it leaves out important details on the actual implementation that makes use of the question-completeness metric to fine-tune it to begin reasoning early.

Let me give an example. In section 3, the authors mentioned that:
> The first scenario includes questions which can be considered “complete" before reaching the end. In such cases, the model can start reasoning early and simply ignore the remaining question. In the second scenario, sufficient information may be available to start reasoning before the question ends, but the model still needs the remaining information to provide a correct response. We propose two different methods to enable early thinking.

This is a great introduction to the motivation behind their approach, and it is very clear. But then the authors proceed by saying:
> To endow the
model with the ability for early reasoning, we created training examples by using our proposed
Question Completeness metric. This metric is designed to identify the optimal time for the model to
begin generating its CoT. Subsequently, we fine-tuned the model on this dataset to teach it to follow
the distribution of these early-reasoning examples.

First, I understand this metric can identify the optimal time to begin generating its CoT and we can create some kind of training dataset from it. But how does this lead to "Subsequently, we fine-tuned the model on this dataset to teach it to follow
the distribution of these early-reasoning examples." I can understand this sentence on its own but I cannot see how the previous metric allows us to "follow the distribution" (this phrase is vague on its own). Even when I read on to section 2.2, I cannot clearly see how we can use the created training examples to "force" the model to start thinking earlier. My very vague guess is that we need to insert the "start_COT" token to the end of the inflection point, which forces the model to start thinking and fine-tune the model from this? In addition to this, we need to append some reasoning traces to the training dataset as well? I might be wrong (and I'm not surprised I am), but that's because the papers leave us to piece together the explanation.

I think there is some logic gap here (arguable at the most important part of the paper) and the authors should elaborate the details better. I'm sure the authors know how the details work but we as readers need to make educated guesses.

**Questions:**

1. Please elaborate how the training dataset can nudge the language model to start reasoning at certain inflection point. if the author can elaborate on this clearly in the rebuttal, I will consider increasing the score.
2. The authors also claimed that their method is Moshi + CoT (ours)♣ and because of this, they outperformed Moshi by a lot (2-3x). This is kind of misleading - I think simply injecting CoT into speech modeling is not that novel and should only serve as a baselines. The authors should focus on the improvement in latency in the approach (by interleaving reasoning while the inputs are still streaming in).
3. Third, I think time is something that is indicative of a good speech model. The authors mostly use number of tokens as an indication of latency but what kind of time are we talking about here? Are we able to reply in a small amount of time?

---

> ### Author Response · Authors · 2025-11-21
> **Response to Reviewer d8Do**
>
> We appreciate the reviewer for recognizing our contribution regarding the Question-Completeness (QC) metric. We are pleased to provide the requested detailed elaboration concerning the LLM fine-tuning process.

---

> ### Author Response · Authors · 2025-11-21
> **Response to Reviewer d8Do Q1 Q2 Q3**
>
> ## Q1: Clarification about the training process and use of the QC metric
>
> > Information not clear for training Moshi to follow the training examples constructed using the proposed QC metric and how does the model initiate early reasoning during inference?
>
>
> ### For training (SFT fine-tuning):
>
> For each training example we have the question $Q$, reasoning $R$ and answer $A$. Using these, we can compute the QC curves (Equation 6) for each training sample, and get the inflection point by setting a defined threshold, say $\theta=0.75$.
>
> Next, we construct the actual training token sequences by shifting the reasoning token text to the inflection point (as shown in Figure 1). The model learns to follow the early thinking distribution by optimizing the next token prediction objective. The model learns the timing of inflection point by optimizing the probability of predicting `<start_cot>` token.
>
> The details about how these token sequences are used to train the model are same as that in the original Moshi paper, and so we have summarized it in Section 2.1.
>
> ### For inference
>
> It is critical to note that *during the inference phase, the inflection point is not explicitly provided to the model.*
>
> To initiate the reasoning, the model samples the `<start_cot>` token conditioned on all past generated tokens and the currently available partial user question.
>
> Consequently, the model must internally predict the inflection point by assessing the evolving semantic content of the partial user query. The entire training methodology is designed to increase the probability of sampling the `<start_cot>` token at the moment the model determines the input is sufficiently complete.
>
> ## Q2: Novelty of the contributions
>
> > The authors also claimed that their method is Moshi + CoT (ours)♣ and because of this, they outperformed Moshi by a lot (2-3x). This is kind of misleading - I think simply injecting CoT into speech modeling is not that novel and should only serve as a baselines. The authors should focus on the improvement in latency in the approach (by interleaving reasoning while the inputs are still streaming in).
>
>
> We respectfully disagree with the reviewer. Our "Moshi + CoT (ours)♣" is critical for the following reasons:
>
> 1. **Novelty in Spoken Reasoning:** To our knowledge, we are the first to investigate the empirical effect of injecting CoT reasoning directly into a full-duplex model for the spoken question-answering task.
> 2. **Streaming ASR Integration:** The proposed architecture includes the design of an internal Streaming ASR component. We are unaware of any prior work that introduces a fully integrated streaming ASR mechanism into a full-duplex model for question answering. The efficacy of this internal streaming ASR is empirically validated in Table 2. The removal of this component results in a substantial 20% degradation in accuracy on the ARC-E benchmark, underscoring its critical role in performance.
> 3. **Deployment Efficiency:** This Streaming ASR is architecturally integrated into a single, unified model. This clever design eliminates the need for an external ASR utility, thereby preventing additional deployment latency and avoiding the limitations associated with multi-component systems.
>
> Secondly, the reviewer's suggestion to "focus on the improvement in latency in the approach (by interleaving reasoning while the inputs are still streaming in)" is in fact the whole idea of "thinking while listening". Our proposed methods -- (i) SFT with the novel QC metric and (ii) DPO for adaptive reasoning -- are explained in detail in Section 3, and the corresponding results are presented in Section 5.2.
>
> ## Q3: Latency of the model
>
> > I think time is something that is indicative of a good speech model. The authors mostly use the number of tokens as an indication of latency but what kind of time are we talking about here? Are we able to reply in a small amount of time?
>
> Our fastest model achieves a response time of 1 second under the default decoding and model deployment configuration. This latency is highly competitive and represents a very small overhead for questions that require complex reasoning. Crucially, this performance can be further accelerated by a factor of up to 3× through careful engineering design and hardware-aware optimization.
>
> In addition, it is also possible to incorporate "thinking while speaking" approaches to fill up the 1 second latency, which reduces the perceived latency by users. We leave this extension for future work.

---

> > ### Comment · Reviewer_d8Do · 2025-11-26
> >
> > Thank you for the prompt response.
> >
> > I mentioned that "Please elaborate how the training dataset can nudge the language model to start reasoning at certain inflection point. if the author can elaborate on this clearly in the rebuttal, I will consider increasing the score." in my initial review. After reading the responses, I am somewhat clearer on the overall approach, but the explanation is quite brief still.
> >
> > I am raising my score to 6 due to the author's comprehensive review. However, I am still on the fence about the clarity of their methods so I cannot increase my score any further.

---

> ### Author Response · Authors · 2025-11-28
>
> We sincerely thank the reviewer for the increased score and for their thoughtful engagement with our work. We appreciate that the reviewer now has a better understanding of our proposed method. However, it seems that the reviewer is not fully clear about how our training dataset nudges the language model to start reasoning at a certain inflection point. To address this concern, we provide a more detailed explanation and a concrete example illustrating the training mechanism.
>
> We use the example in Figure 1 to give a more detailed explanation. Although this example is more of a factual question that doesn't require extensive reasoning, its brevity makes it ideal for clearly demonstrating the fundamental structure of our approach.
>
> The question, reasoning trace and answer are:
> * Question ($Q$): What is the capital of France? Is it Barcelona, London, or Paris?
> * Reasoning ($R$): The captial of France is Paris.
> * Answer ($A$): It is Paris
>
> ## Training Stage
>
> Suppose we get the inflection point $\zeta$ at the word `France` in the user question by 95\% threshold. (Notice that the QC values are calculated based on $Q$,$R$, and $A$)
>
> We then constructed the input for our model as the following example: (split into two parts due to line width limit)
> ```
> User Audio Stream   : What       is      the     capital     of     France?       Is  it     Barcelona,
> Text Monologue      : ~~~~~~What       is      the     capital     of     France?(The)  Is  it     Barc
> System Audio Stream : [Silence-------------------------------------------------------------------------
> ```
> ```
> User Audio Stream   :          London,    or  Paris?[Silence-------------------------------------]
> Text Monologue      : elona,(captial)London,(of)or  Paris?(of France is Paris)It   is     Paris.
> System Audio Stream : -------------------------------------------------------] It   is     Paris.
> ```
> Notations in the example:
> * `~~~~~~`: constant delay for the User ASR
> * `[Silence---]`: silence codec sequence (we obtain the codec sequence by inputting silence waveform into the Mimi codec)
> * Words in parentheses `(`, `)` on the text monologue stream: reasoning tokens
> * Words without parentheses in the text monologue stream: aligned transcript either for User Audio Stream or System Audio Stream.
>
> To improve readability and focus on the overall token arrangement, we omit special tokens in the example. In practice, to allow interleaving between streaming ASR and reasoning tokens, we use `<switch_cot>` and `<switch_asr>` tokens to denote the changes between two modes.
>
> As shown in this example, the user streaming ASR tokens in the text monologue starts after a constant delay frames.
> The streaming ASR tokens are time-aligned with the user audio stream.
>
> Given that the inflection point is the word `France`, we start interleaving the reasoning trace after the word `France` is spoken and transcribed in our streaming user ASR. Hence, the first word of the reasoning trace `(The)` is inserted right after the transcript `France` in the text monologue. Once the reasoning trace started, we interleave the remaining reasoning tokens between the user streaming ASR tokens if possible. For shorter words (e.g., is, it...), we do not interleave reasoning tokens due to space limitations.
>
> After the reasoning trace ends, we insert the system audio transcript in the text monologue for the model to output the answer response. The duration of silence is calculated to make all 3 channels have the same temporal length.
>
> For training, the model inputs are in the shape of [$17 \times T$]
> For the user and system audio channel, each of them is encoded using 8 codebooks, and there is only one codebook for the text monologue. Hence, the overall dimension is $8 + 8 + 1 = 17$
>
> The training objective is a simple next token prediction loss on System Audio and text monologue ($9 \times T$). Under this objective, the model learns the distribution of the token sequence in the text monologue.
>
> Back to the question:
> > how the training dataset can nudge the language model to start reasoning at certain inflection point.
>
> The text monologue token sequence in our training dataset is constructed based on the inflection point. **Specifically, the start of the first reasoning token is the inflection point**. As a result, by finetuning the model on such a training sequence, the model learns the pattern of the first reasoning token, which is essentially the inflection point.

---

> > ### Author Response · Authors · 2025-11-28
> >
> > ## Inference Stage
> > During inference, only the User Audio Stream is given. The user streaming ASR, reasoning trace, system audio, and QC values/inflection point are not given.
> >
> > Let's break down step by step the generation process.
> >
> > For the first few frames(same as the number of delayed frames in our training setting), we forced decode null tokens (~) for the purpose of streaming ASR.
> > ```
> > User Audio Stream   : What
> > Text Monologue      : ~~~~~~
> > System Audio Stream : [Silence
> > ```
> >
> > After that, the model decodes freely, and it starts to do streaming ASR in the text monologue.
> >
> > ```
> > User Audio Stream   : What       is      the     capital     of     France?
> > Text Monologue      : ~~~~~~What       is      the     capital     of     France?
> > System Audio Stream : [Silence---------------------------------------------------
> > ```
> >
> > At each decoding step, other than decoding normal text tokens, the model is also able to sample special tokens. Specifically, if the model sampled `<switch_cot>` followed by `<start_cot>` at a certain step after sampling streaming user ASR tokens, it triggers the reasoning trace generation. **This indicates that the model predicts the inflection point has arrived, and thus it starts to generate the reasoning trace.** If the model learns the training data well, the timing of sampling the first `<switch_cot>` will be similar to that in the training data, which also implies that the model learns to predict the inflection point demonstrated in the training dataset well.
> >
> > ```
> > User Audio Stream   : What       is      the     capital     of     France?
> > Text Monologue      : ~~~~~~What       is      the     capital     of     France?(
> > System Audio Stream : [Silence----------------------------------------------------
> > ```
> > The parentheses `(` as an indication of the model sampling `<switch_cot>` followed by `<start_cot>`.
> >
> > Afterward, the model freely generates the remaining tokens. We stop the generation when the `<finish_answering>` is sampled in the text monologue.

---

> > > ### Author Response · Authors · 2025-11-28
> > >
> > > ## Conclusion
> > > In conclusion, our method explicitly links the start of reasoning to the structure of the training data. Specifically, we construct the training sequences such that the first appearance of the `<switch_cot>` token (immediately followed by the ``<start_cot>`` token) precisely marks the timing of the inflection point.
> > >
> > > By consistently training the model on this pattern, we compel it to learn the association between the input context and this specific special token sequence. This effectively equates to teaching the model the optimal timing to initiate the reasoning process at the defined inflection point.
> > >
> > > ## Conclusion for reviewer d8Do's comment
> > > To ensure complete clarity for all readers, we commit to including expanded training and inference details within the revised manuscript. This will enable a deeper understanding of our approach's implementation.
> > >
> > > We believe the detailed explanation and illustrative example provided above fully clarify the mechanism of how we teach our model to start reasoning at a given inflection point.
> > >
> > > We thank the reviewer once again for their constructive feedback and are happy to provide any further clarification if there is any certain aspect that is still unclear to the reviewer.

---

### Official Review · Reviewer_DB5G · 2025-10-26

**Soundness:** 3
**Presentation:** 3
**Contribution:** 3
**Rating:** 6
**Confidence:** 4

**Summary:**

This paper presents a Speech Language Model (SLM) that enables "thinking while listening." The model is finetuned on Moshi and introduces a question completeness metric to support early reasoning when only partial user queries have been received. Additionally, the model leverages Direct Preference Optimization (DPO) to further enhance its performance.

**Strengths:**

- The idea of enabling "thinking while listening" in SLMs is novel.
- Experimental results demonstrate strong performance for the proposed model.
- Comprehensive ablation studies convincingly show the effectiveness of each proposed module.

**Weaknesses:**

- The paper does not evaluate end-to-end latency in terms of seconds (since the streaming ASR and the output of reasoning tokens may introduce further latencies). Latency is a critical factor for achieving natural conversational experiences, especially for full-duplex SLMs such as Moshi.
- Training relies solely on synthetic audio, limiting the model's ability to handle diverse input voices. Although not a critical weakness, this raises concerns about potential overfitting to the voice characteristics of the training speech. It would be valuable to assess the model's robustness on varied input voices. The authors could consider benchmarks like [VoxEval, 2025] (support both mathematical reasoning and diverse input audio conditions), [VoiceBench, 2024] (support diverse input audio conditions), or similar alternatives.

**Questions:**

- Does the latency discussed in this paper refer to reasoning token length or answer token length? This differs from the "first package latency" commonly referenced in SLM systems. The distinction should be clearly explained in the abstract and introduction to avoid confusion.
- Is the streaming ASR naturally supported by the Moshi model, or is it trained by the authors?
- How much latency is added compared to the original Moshi model? Is it just the 480ms? Can you count all the latencies, e.g., Codec encoding, 480ms, LLM inference, and Codec decoding?
- In the speech-based CoT setting, the authors appear to prefix the speech CoT sequence to the spoken response. Why not allow the model to generate the speech CoT reasoning sequence itself? A fairer comparison between Text CoT and Speech CoT would involve the model autonomously generating the Speech CoT.

---

> ### Author Response · Authors · 2025-11-21
> **Response to Reviewer DB5G**
>
> We thank Reviewer DB5G for recognizing the novelty of our "thinking while listening" paradigm and for the positive acknowledgment of our strong performance and comprehensive ablation studies. We appreciate the engagement with our work; however, we would like to offer clarification regarding several points raised in the review.

---

> ### Author Response · Authors · 2025-11-21
> **Response to Reviewer DB5G Q1**
>
> ## Q1: Clarification about "latency" of the model
>
> In all reported experiments, latency is defined as the time interval (measured in terms of number of tokens) from the end of the user's question to the initiation of the model's spoken response. This metric inherently accounts for the delays associated with both the streaming ASR and the generation of CoT reasoning tokens.
>
> > The paper does not evaluate end-to-end latency in terms of seconds (since the streaming ASR and the output of reasoning tokens may introduce further latencies)
>
> > Does the latency discussed in this paper refer to reasoning token length or answer token length? This differs from the "first package latency" commonly referenced in SLM systems. The distinction should be clearly explained in the abstract and introduction to avoid confusion.
>
> In real-time SLMs like Moshi, decoding happens "on-the-clock": within the span of a fixed time interval (e.g., 80ms), the following events happen: an 80ms chunk of audio is read from the user channel buffer, it is encoded into an audio token and fed into the LLM which generates a system audio token and system text, and the system audio token is decoded into an 80ms waveform. All of these happen within the the fixed interval to ensure there is no accumulation of audio in the input buffer, such that the model is "real-time". In this setting, the wall clock response latency only depends on the gap (or number of tokens) between the last user question token and first system response token, and in fact, the wall-clock latency of our system would be exactly 80ms times the number of tokens we have reported. We chose to report latency in terms of number of tokens since the 80ms multiplier is configurable based on the speech tokenizer used, but the number of tokens actually represents how much the model's reasoning affects time to first packet.
>
> Additionally, it is possible to further speed up the decoding of CoT text tokens depending on different implementations of model serving and hardware constraints. For example, we tested Moshi model on A100 GPU and find out that it takes roughly 26ms to do one forward pass. Hence, it is possible to decode 3 (80ms/26ms) text tokens along with 1 audio token to speed up the waiting time for CoT generation. Reporting latency in token length provides a **hardware and implementation agnostic metric** that ensures a fair and consistent comparison between our proposed model and the original Moshi baseline.
>
>
> > How much latency is added compared to the original Moshi model? Is it just the 480ms? Can you count all the latencies, e.g., Codec encoding, 480ms, LLM inference, and Codec decoding?
>
> Under default decoding and deployment of Moshi (and our model). To convert token length into seconds, we just need to multiply token length by 80ms. Our fastest model with DPO training is able to respond roughly **1 second** on average. The original Moshi responds quickly (about 80ms). We note that we can teach the model to "mask" even this small latency by complementary methods like STITCH, but this is outside the scope of this work.
>
> Since the entire system is architecturally designed to operate in a streaming manner, all component-specific latencies—including those arising from the codec encoder, codec decoder, and other essential processing steps—can be contained and executed within a single Mimi codec token time span of 80ms.

---

> > ### Comment · Reviewer_DB5G · 2025-11-23
> >
> > I appreciate the authors' response regarding latency, and the explanation of "multiply by 80ms" is understandable. However, the authors still do not evaluate end-to-end latency in terms of seconds, which is crucial for readers to quickly grasp the overall latency of the proposed system. Additionally, the authors’ statement—"Our fastest model with DPO training is able to respond roughly 1 second on average"—is not sufficiently professional, as it does not provide a precise value or explain the experimental settings under which this number was obtained.
> >
> > Referring to Table 4, the number of additional tokens in the DPO setting can be as high as 48, corresponding to an additional latency of 3.84 seconds. Furthermore, the accuracy improvement for GSM8K is not significant enough. Therefore, while the paper proposes an interesting approach, the overall results are not particularly compelling. I decide to maintain my score for now.

---

> > > ### Author Response · Authors · 2025-11-24
> > >
> > > We sincerely thank the reviewer for the prompt response and feedbacks. Below, we provide a detailed explanation and clarification regarding the remaining concerns raised.
> > >
> > > > However, the authors still do not evaluate end-to-end latency in terms of seconds, which is crucial for readers to quickly grasp the overall latency of the proposed system.
> > >
> > > We believe there may still be a misunderstanding regarding the end-to-end latency calculation. The end-to-end latency is **measured** in terms of *seconds* but is **reported** in terms of *tokens* in the paper.
> > >
> > > To measure the response latency, we first feed the evaluation user audio (question) into our model and save the entire response as an audio file.
> > > Then we apply Pyannote VAD (mentioned on line 350) on the resulting audio file. This tool directly measures the end-to-end latency in seconds from the end of the user's question to the start of the model's audible response.
> > > The latency value obtained from the VAD tool is a raw time measurement (in seconds). To enable a hardware- and infrastructure-agnostic comparison (the reason is mentioned in previous rebuttal comment), we divide the measured latency by 80 ms to convert it into numbers of tokens.
> > >
> > > We sincerely apologize for the initial lack of clarity regarding the detailed implementation of our latency calculation. We hope the detailed explanation provided in the preceding points has now fully clarified our measurement.
> > >
> > > To ensure this critical detail is transparent to all readers, we commit to the following revisions in the final manuscript:
> > > * We will add a dedicated description in Section 4.2 Evaluation for the procedure of latency measurement using VAD took.
> > > * We will explicitly describe the relationship between seconds and the number of tokens to allow readers to quickly and correctly interpret the system's overall latency.
> > >
> > > > Additionally, the authors’ statement—"Our fastest model with DPO training is able to respond roughly 1 second on average"—is not sufficiently professional, as it does not provide a precise value or explain the experimental settings under which this number was obtained.
> > >
> > > We apologize for the ambiguity. We are referring to the results presented in Table 4. Specifically, on the ARC-E evaluation set, the average reported latency for the DPO setting is 12.0 number of tokens (0.96 sec). The latency calculation is the same for all experiment and is described in the preceding paragraph.
> > >
> > > > Referring to Table 4, the number of additional tokens in the DPO setting can be as high as 48, corresponding to an additional latency of 3.84 seconds. Furthermore, the accuracy improvement for GSM8K is not significant enough.
> > >
> > > Regarding Table 4, we want to first highlight the **significant latency reduction** achieved by our Length-DPO finetuning across all eval sets. For example, on SIQA, we get a **74.2\%** latency reduction. Furthermore, for all eval sets (except GSM8K), the resulting latency is shorter than **1.672** seconds (20.9 tokens×0.08 s/token). This demonstrates effective latency reduction for general reasoning and non-reasoning conversational tasks.
> > >
> > > For the GSM8K mathematical reasoning task, while the latency is higher than other sets due to the inherently greater complexity of math questions, the improvement is still substantial: we reduced the latency from 6.08 seconds down to 3.88 seconds, indicating a remarkable **36\%** reduction.
> > >
> > > We agree that there remains space for further latency improvement, especially before deploying this system directly to real-world users. We can achieve this through two primary directions (also mentioned in previous rebuttal comments)
> > >
> > > * Decoding Optimization: Implementing a better decoding method could potentially yield a 3x speedup. Applying this to the GSM8K result would further reduce the latency down to 1.29 seconds (3.88 / 3 = 1.29).
> > >
> > > * Perceptual Latency Reduction: We can employ concurrent methods, such as STITCH or similar techniques, to allow Moshi to begin speaking during this thinking latency.
> > >
> > >
> > > As for the accuracy improvement, we are not entirely certain which baseline the reviewer is comparing against. We believe there are two likely comparisons the reviewer may be referring to:
> > >
> > > * Comparison between SFT and DPO results in Table 4:
> > >
> > >   We would like to emphasize that the primary goal of Length-DPO is not to achieve a new SoTA in accuracy, but rather to significantly shorten the end-to-end latency while effectively maintaining the accuracy achieved by SFT. As shown in Table 4, the results across all eval tasks confirm the efficacy of this approach: the accuracy is robustly maintained while the end-to-end latency is significantly improved.
> > > * Comparison between DPO result in Table 4 and the original Moshi Model:
> > >
> > >   We get an accuracy of 14.7\% for Length-DPO (Table 4) on GSM8K while the original Moshi gets an accuracy of 8.7\% (Table 2). This 6\% improvement is quite significant,  especially for GSM8K math reasoning task.

---

> ### Author Response · Authors · 2025-11-21
> **Response to Reviewer DB5G Q2**
>
> ## Q2: Synthetic v/s real audio
>
> > Training relies on solely synthetic audio, limiting the model’s ability to handle diverse inputs. Although not a critical weakness, this raises concerns about potential overfitting to the voice characteristics of the training speech.
>
> We acknowledge the reviewer's concern regarding the current model's acoustic robustness. While this represents a valuable area for improvement, we believe that acoustic generalization diverges from the primary focus of our work. The fundamental motivation and novelty of our paper are centered on the following three points:
>
> * The design of a novel mechanism for injecting CoT reasoning into the Moshi model together with streaming ASR.
> * The development of the QC metric for indicating optimal timing for early reasoning.
> * The DPO finetuning for boosting performance of early reasoning.
>
> The issue of acoustic generalization can be effectively addressed in future work by:
>
> * Data augmentation: Incorporating randomized noise conditions and other noise augmentation techniques during the training phase.
> * Curating real word data for fine-tuning: Curating a small, high-quality dataset of human-spoken QA data specifically for targeted fine-tuning.

---

> ### Author Response · Authors · 2025-11-21
> **Response to Reviewer DB5G Q3**
>
> ## Q3: Other questions
>
> > Is the streaming ASR naturally supported by the Moshi model, or is it trained by the authors?
>
> The streaming ASR design is fully trained by us. We emphasize that the design of this internal streaming user ASR constitutes a crucial contribution for the following three reasons:
>
>
> 1. **Novelty**: To our knowledge, this represents the first example of integrating a fully streaming ASR system directly into a full-duplex model designed for question answering.
> 2. **Effectiveness**: The efficacy of this design is clearly demonstrated in Table 2. Notably, removing the internal streaming ASR results in a significant 20% accuracy degradation on the ARC-E benchmark, underscoring its performance impact.
> 3. **Efficiency**: This approach cleverly integrates the user streaming ASR into a single, unified model. This eliminates the need for an external ASR tool, thereby avoiding additional deployment latency and external dependencies.
>
>
> > In the speech-based CoT setting, the authors appear to prefix the speech CoT sequence to the spoken response. Why not allow the model to generate the speech CoT reasoning sequence itself? A fairer comparison between Text CoT and Speech CoT would involve the model autonomously generating the Speech CoT.
>
> We believe there is a misunderstanding regarding the experimental setup for text and speech based CoT. In the speech-based CoT setting, the model autonomously generates the spoken CoT sequence itself during inference, ensuring a fair and apples-to-apples comparison with the text-based CoT setting.
>
> To provide explicit clarity on the Speech-based CoT implementation:
>
> * Training Phase: As detailed on Line 407, the Moshi model was fine-tuned using training data where the spoken CoT was prepended to the final spoken answer. This teaches the model to speak the reasoning in spoken format.
> * Inference Phase: The model receives the user's spoken question and is then responsible for independently generating the entire output sequence. This generated output sequence explicitly includes spoken CoT followed by the final answer. Crucially, the ground-truth spoken CoT is **not provided** to the model at inference time.
>
> Therefore, in both the Text CoT and Speech CoT experimental settings, the models are required to autonomously generate their own reasoning trace during the evaluation phase. Hence, it is a fair comparison.

---

### Official Review · Reviewer_8SXS · 2025-10-27

**Soundness:** 3
**Presentation:** 3
**Contribution:** 2
**Rating:** 4
**Confidence:** 4

**Summary:**

This study presents the following key contributions:

1. It extends the existing full-duplex SLM, *Moshi*, by integrating a streaming ASR component into its text channel.
2. It incorporates the reasoning process while introducing a dedicated mechanism to determine the appropriate timing for reasoning during a user’s utterance, thereby aiming to reduce latency.
3. Furthermore, it employs Direct Preference Optimization (DPO) to guide both response length and answer correctness.

Through these methods, the authors improve the overall performance of the conversational model and demonstrate that by adjusting the threshold related to reasoning timing, one can effectively balance and trade off between latency and performance.

**Strengths:**

1. The paper is well written overall and effectively uses figures to support understanding.
2. The authors successfully demonstrate the effectiveness of the introduced ASR modeling component and the use of DPO.
3. They also show that latency can be reduced through appropriate adjustment of the threshold.

**Weaknesses:**

I have some fundamental questions regarding the underlying assumption of the scenario proposed in this paper. In typical real-world interactions with voice assistants, I am not sure how frequently situations arise in which it is actually reasonable or necessary for the model to perform reasoning *during* the user’s utterance.

More specifically, while it seems plausible for the model to reason incrementally based on partial chunks of the user’s speech, I wonder how common it really is for a partial question and the entire question to yield essentially the same reasoning trajectory. In addition, since a 95% threshold would likely result in only a very small difference in word count compared to the full question, I remain somewhat skeptical about whether it is actually possible to reduce latency meaningfully without degrading performance.

As the authors also mention, lowering the threshold to around 0.65 in order to achieve noticeable latency improvement leads to significant performance degradation. This makes me wonder whether, in most cases, the positions where the model determines that it can safely perform reasoning, say, around the 95% mark of the utterance, are in fact not meaningfully different from having access to the full question.

Even in the figures provided by the authors, except for a few special cases, the reasoning appears to begin near the final 3~5 tokens of the input. Since this portion is part of the prefill stage, I am uncertain whether this actually produces a substantial latency advantage. Consequently, I find myself questioning the necessity and justification of this specific module and setup.

Perhaps I overlooked it, but I would like to know, based on standard speech-related benchmarks (e.g., *VoiceBench*), approximately how many of the final words are typically excluded on average when using a 95% threshold.

To be clear, my comments are not meant to question the overall necessity of the task itself, but rather to express skepticism about whether the proposed setup reflects a truly *practical* real-world scenario. I hope this perspective is taken in that spirit.

**Questions:**

The only question I raised is addressed in the *Weaknesses* section, and there is a citation typo on line 348 (a missing parenthesis).

---

> ### Author Response · Authors · 2025-11-21
> **Response to Reviewer 8SXS**
>
> We thank Reviewer 8SXS for the constructive feedback and for noting the strong performance achieved by incorporating streaming ASR and DPO fine-tuning. The main points of concern, relating to the experimental setup, are addressed in detail below.

---

> ### Author Response · Authors · 2025-11-21
> **Response to Reviewer 8SXS Q1 (1/2)**
>
> ## Q1: Is it necessary and/or reasonable for the model to perform reasoning during the user's speech?
> > I have some fundamental questions regarding the underlying assumption of the scenario proposed in this paper. In typical real-world interactions with voice assistants, I am not sure how frequently situations arise in which it is actually reasonable or necessary for the model to perform reasoning during the user’s utterance.
>
> Let's decouple the questions:
> 1. Is it reasonable or necessary for voice assistants to perform reasoning?
> 2. Is it reasonable or necessary to perform reasoning during the user's utterance?
>
> We believe that reasoning is crucial to improve response accuracy under the current regime of autoregressive models. Evidence for this is also shown in Section 5.1, where models with CoT-based reasoning show 2-3x better accuracy compared to those that do not generate any reasoning traces.
>
> Now, if we agree that reasoning is advantageous for the model, can it be performed during the user's utterance? Before answering this, let's answer an even simpler question: *why would we need to start reasoning early?*
>
> Arguably, the most important metrics to track for any voice assistant are: (i) accuracy of the response, and (ii) time to task completion (i.e., how long did the model take to get to the correct answer). While CoT improves (i), it may make the response extremely long and laborious (especially for voice-based models), leading to significant increase in time to task completion. To tackle this, we propose reasoning in text, since a sequence of text tokens is much shorter than a sequence of audio tokens conveying the same lexical content. Table 3 (Section 5.1) shows that this modality switch even retains model accuracy. However, it introduces a new problem: the user-perceived latency (UPL) increases since there is no spoken output corresponding to the reasoning.
>
> To address this new problem of UPL, we can either teach the model to "talk" while reasoning (as done in concurrent work such as STITCH), or to start the reasoning earlier (as we do). Notably, both of these are complementary methods, with the same goal to fill the void caused by reasoning.
>
> Let us now come to the final question: *is it even possible to reason early?* On our CoT-collection validation set, with a fairly conservative threshold of QC = 95%, we have the following distributions.
> About 70% of the data is possible to reason early under QC=95% and we draw the distribution of these data.
>
> \# of Left Shifted User Words:
> ```
> Max: 46, Min:1, Mean: 3.76
> Percentage (%) | Bin Range (# of Words)   | Histogram
> -------------------------------------------------------
>  20.50%        | [1.00, 1.90)              | ████████████████
>  24.65%        | [1.90, 2.80)              | ████████████████████
>  16.23%        | [2.80, 3.70)              | █████████████
>   9.50%        | [3.70, 4.60)              | ███████
>   7.78%        | [4.60, 5.50)              | ██████
>   5.39%        | [5.50, 6.40)              | ████
>   4.83%        | [6.40, 7.30)              | ███
>   3.31%        | [7.30, 8.20)              | ██
>   2.57%        | [8.20, 9.10)              | ██
>   1.52%        | [10.00, 10.90)            | █
>   1.17%        | [10.90, 11.80)            |
>   (tail truncated)
> ```
>
> \# of Left Shifted Time (Second):
> ```
> Max: 28.48, Min:0.08, Mean: 1.27
> Percentage (%)  | Bin Range (Seconds)      | Histogram
> -------------------------------------------------------
>  38.58%        | [0.08, 0.65)              | ████████████████████
>  25.04%        | [0.65, 1.22)              | ████████████
>  13.56%        | [1.22, 1.78)              | ███████
>   9.64%        | [1.78, 2.35)              | ████
>   5.47%        | [2.35, 2.92)              | ██
>   3.33%        | [2.92, 3.49)              | █
>   1.69%        | [3.49, 4.06)              |
>   (tail truncated)
> -------------------------------------------------------
> ```
>
> On average, we can start reasoning **1.27 seconds** early, which corresponds to ~**16 text tokens** with our vocabulary. This number may be as high as ~**31 tokens** in certain cases (often for multiple-choice style questions). We hope this answers the reviewer's concern about the *feasibility* of early reasoning.

---

> ### Author Response · Authors · 2025-11-21
> **Response to Reviewer 8SXS Q1 (2/2)**
>
> > Perhaps I overlooked it, but I would like to know, based on standard speech-related benchmarks (e.g., VoiceBench), approximately how many of the final words are typically excluded on average when using a 95% threshold.
>
> The majority of prior benchmarks (such as VoiceBench) are not suited for our work since they only evaluate factuality rather than reasoning problems.
>
> Nevertheless, the CoT-Collection dataset (which we use for train and validation) contains ~25k **OpenBookQA** samples which are also included in VoiceBench. We computed some statistics for the number of words excluded when using a 95% threshold on this set, as shown below:
>
> |                                   | Average | Min. | Max. |
> |-----------------------------------|---------|------|------|
> | # of words before end of question |  2.63       | 0      |  35    |
> | Time duration (Sec)                     |    0.97     | 0     | 10.64     |
> | Number of reasoning tokens        |  12.07       |     0 |  133    |
>
> Clearly, this eval set also allows the model to emit a non-negligible number of reasoning tokens before the end of user's question, despite using a relatively conservative threshold of 95%.
>
> In Section 3.2, we have also shown results using **LLaMA-QS**. The questions in this dataset are characterized by their brevity, conciseness, and predominantly factual nature, closely mirroring the common user questions found in VoiceBench and other established voice assistant benchmarks. This is a suitable proxy for evaluating performance on this class of short, non-reasoning queries.

---

> ### Author Response · Authors · 2025-11-21
> **Response to Reviewer 8SXS Q2**
>
> ## Q2: Is it common for a partial question and the entire question to yield essentially the same reasoning trajectory?
>
> > More specifically, while it seems plausible for the model to reason incrementally based on partial chunks of the user’s speech, I wonder how common it really is for a partial question and the entire question to yield essentially the same reasoning trajectory.
>
> It is NOT, and this is primary motivation for our second contribution: preference tuning with examples created using rejection sampling. We call it "correctness-based preference" in our paper, and the summary of the approach is the following:
>
> 1. We first use SFT to train a base model to start reasoning early, say at QC=75%.
> 2. We use the SFT model to generate $K$ responses (reasoning + answer) per prompt for a subset of the training data.
> 3. From among these $K$ responses, we create a training tuples (prompt, +ve, -ve), where the +ve and -ve examples are where the model starts reasoning at roughly the same time, but the +ve leads to a correct answer whereas the -ve does not.
>
> By designing the preference data in this way, we **teach the model to update its reasoning trace based on new information**, i.e., not just rely on the history. An example of this scenario is shown in Table 8 (Example 3) in Appendix A.5. In the QC($\theta$=0.75) + CorrectDPO setting, the CoT starts before the final option "job" is spoken by the user. By DPO finetuning, our model is still able to take the incoming information into consideration and thus get the correct answer whereas the SFT model fails.
>
> Overall, this method is extremely effective: from Fig. 5 (purple pentagon curve), we see that Correct-DPO provides 10-15% relative accuracy improvements over the base model without significant delay in start of reasoning.
>
> > As the authors also mention, lowering the threshold to around 0.65 in order to achieve noticeable latency improvement leads to significant performance degradation. This makes me wonder whether, in most cases, the positions where the model determines that it can safely perform reasoning, say, around the 95% mark of the utterance, are in fact not meaningfully different from having access to the full question.
>
> Our work in the paper is only the first to perform early reasoning, and we believe that the degradation from early reasoning can be improved with better techniques for updating the reasoning trace with incoming information.
>
> At a high level, a model that waits until the end of the user's question to start thinking would need to attend to the entire audio history to reason and respond correctly. This is like trying to answer a very long question purely based on your memory. On the other hand, letting the model start reasoning during the user's question is like **providing it a scratch space to make notes** of the important information. Once the question ends, it can then attend more strongly to these notes. We conjecture that this is also the reason why simply adding streaming ASR to the text monologue channel improves the response accuracy significantly (for example, 55.8% to 77.7% on ARC-E, as shown in Table 2).

---

> ### Author Response · Authors · 2025-11-21
> **Response to Reviewer 8SXS Q3**
>
> ## Q3: Does the method result in any substantial latency advantage?
>
> > Even in the figures provided by the authors, except for a few special cases, the reasoning appears to begin near the final 3~5 tokens of the input.
>
> This observation may be due to a confusion between written v/s spoken words. From the plots we showed above, we see that the model can start reasoning approximately **3-5 "written" words** before the end of the user's utterance. However, since these words are actually spoken out loud, they **span ~1-2 seconds**, based on an average conversational speaking rate of 120-150 wpm. Since our model emits 1 token per 80ms of audio, it can consequently **generate 25 reasoning tokens** by starting just 2 seconds before the end of user's question.
>
> > Since this portion is part of the prefill stage, I am uncertain whether this actually produces a substantial latency advantage.
>
> We are not quite sure what the reviewer means by "prefill stage". Under the default decoding configuration, the Moshi backbone operates in a synchronous streaming manner on the clock: an 80ms audio chunk comes in and is encoded into 1 user token; this triggers the concurrent generation of one system audio token and one system text token. Critically, text is a more highly compressed medium than its audio representation. As a result, a shift corresponding to one user spoken word is roughly equivalent to 4–5 text words of space, given that our text tokenizer generally maps one word to one token.
>
> As a result, the percentage of examples that benefit from early thinking under QC 95% threshold is quite frequent and the latency reduced is very meaningful.

---

> > ### Comment · Reviewer_8SXS · 2025-11-27
> >
> > Thank you for your clarification, I'll modify my score accordingly.

---

> > > ### Author Response · Authors · 2025-11-27
> > >
> > > We sincerely thank the reviewer for the increased score and for their constructive feedback throughout this process.

---

### Author Response · Authors · 2025-11-30
**Summary of Rebuttal Process**

We sincerely thank the AC, PC, and all reviewers for their substantial time and effort. Their constructive suggestions and insightful comments were invaluable in strengthening our paper. In this short note, we want to summarize the overall discussions that happened before the early termination of the review period.

## Summary of Strengths
The 3 reviewers highlighted multiple strengths of our paper, which can be summarized in three key aspects:

1. Novelty and Relevance: (Reviewer d8Do, DB5G)

    The core idea of enabling "thinking while listening" in Speech Language Models (SLMs) is both novel and highly relevant/practical.
2. Strong Performance and Effectiveness (Reviewer 8SXS, DB5G)

    * **Question Completeness(QC) metric makes perfect sense**, which allows us to identify when the LLM has enough information to start thinking about the answer. *Reviewer d8Do specifically marked this contribution as "excellent."*
    * Comprehensive experiments have shown the effectivenes of each proposed module (streaming ASR, DPO finetuning, QC metric) in our work
    * Latency reduction: Effective latency reduction through appropriate adjustment of the threshold.
4. High Quality of Writing (Reviewer 8SXS, d8Do)
## Summary of Reviewers' Concerns
### Reviewer 8SXS:

1. *Is it necessary and/or reasonable for the model to perform reasoning during the user's speech?*

    Early reasoning during user's speech is both necessary and reasonable, a conclusion supported by statistics demonstrating 'thinking early' patterns in our dataset.
2.  *Is it common for a partial question and the entire question to yield essentially the same reasoning trajectory?*

    No, partial and complete questions often yield different reasoning traces, a finding that provides motivation of our DPO fine-tuning approach. We further substantiate this claim with an illustrative example detailed in the Appendix.
3.  *Does the method result in any substantial latency advantage?*

    The reviewer has confusion between written v/s spoken words. We clarify that by starting the reasoning 3-5 spoken words early, we are able to generate 13-25 reasoning tokens earlier.

*Response after reading our comments*

We successfully addressed these concerns, which led the reviewer to provide positive feedback and **increase their rating** (initial score: 4).

### Reviewer 8SXS:

1. *Concern about our model's acoustic robustness due to synthetic training data.*

    We acknowledge the reviewer's concern, but we note that addressing this specific issue **falls outside the scope of the present work**. Nonetheless, we have outlined several promising directions for future mitigation, including the use of data augmentation and the curation of real-world data for fine-tuning.
2. *Clarification about "latency" of the model.*

    We provided a detailed description of our model's streaming inference process and clarify that the use of token count as the unit for latency measurement is hardware- and implementation-agnostic. Furthermore, this doesn't undermine the fair comparison against the baseline model (Original Moshi).

*Response after reading our comments*

While we address most of the reviewers' concerns, there is still a misunderstanding about our latency measurement. The reviewer mistakenly thought that we calculate the latency by counting the #tokens. In fact, we **measure** the end-to-end latency with VAD tool in seconds but **reported** as #tokens in the paper. However, the reviewer hasn't responded to our latest comments.

### Reviewer d8Do
1. *Need clarification about our fine-tuning and inference process.*

    We provide a detailed explanation about how we construct our training examples based on the inflection point and mention that during inference, the models are able to predict the timing to start reasoning without the inflection point provided.

*Response after reading our comments*

We received positive feedback and an **increased rating** after our initial explanation (initial score: 4), but the reviewer still remains a little unclear. Although the reviewer does not point out a specific aspect that is unclear, we provide a concrete example to show our training and inference approach and emphasize that
> The position of `<switch_cot>` followed by  `<start_cot>` token corresponds to the inflection point derived by our QC metric. (We construct the training examples this way.) Learning the position of these special tokens is the same as learning the inflection point.

We think that this should sufficiently address the remaining concern.

---

### Author Response · Authors · 2025-11-30
**Concluding Remarks by the Authors**

Overall, the reviewers recognized the novelty, contributions, and strength of our results. The initial concerns largely stemmed from a misunderstanding of some concepts within the paper. **Following our initial rebuttal, two out of three reviewers favorably increased their scores**. **The two reviewers that increased the scores give us weak reject initially and the change of scores happened before the API incident.** While we subsequently provided responses addressing their remaining questions, we were unable to receive further feedback due to the early termination of the review period.

We are committed to revising our paper by incorporating the explanations provided during this rebuttal process. These revisions will directly address the points raised by the reviewers and will focus on improving clarity for the reader. Specifically, we will add expanded content detailing:

* The methodology for measuring and reporting latency in our experiments.
* A reporting of latency results in seconds to provide a real-world metric.
* A thorough explanation of the inflection point's connection to both the training and inference pipelines.

These additions will ensure that our results and methods are easier to understand and evaluate.

We put the summary of the rebuttal period in the comment "Summary of Rebuttal Process" due to character limit.

---

### Meta-Review · Area_Chair_X74Y · 2026-01-07

**Summary:**

This paper received 4,6,4 scores originally. The comments from the first reviewer is about latency. Authors did a good job addressing it. R1 mentioned that they will modify the score accordingly. AC takes it as improving the score. On the other hand, R3 thinks the novelty is limited and didn't have the opportunity to consider authors' feedback, so this depends on AC's perception. AC read the paper and thinks that this paper is at least useful, although it might not look very novel. Being useful is a good thing. AC thinks about this paper for a long time and thinks that this work will be a good one for the audio community and would like to recommend accept.

**Reviewer Concerns:**

See above.

**Reviewer Scores:**

See above.

---

### Decision · Program_Chairs · 2026-01-26

Accept (Poster)